# Importance of snowmelt contribution to seasonal runoff and summer low flows in Czechia

Michal Jenicek[1], Ondrej Ledvinka[2]

[1]Charles University, Department of Physical Geography and Geoecology, Prague, Czechia
[2]Czech Hydrometeorological Institute, Prague, Czechia

*Correspondence to*: Michal Jenicek (michal.jenicek@natur.cuni.cz)

**Keywords.** snow, snowmelt runoff, recharge, baseflow, low flow

**Abstract.** The streamflow seasonality in mountain catchments is often influenced by snow. However, a shift from snowfall to rain is expected in the future. Consequently, a decrease in snow storage and earlier snowmelt is predicted, which will cause changes not only in seasonal runoff distribution in snow-dominated catchments, but it may also affect the total annual runoff. The objectives of this study were to quantify 1) how inter-annual variations in snow storages affect spring and summer runoff, including summer low flows and 2) the importance of snowmelt in generating runoff compared to rainfall. The snow storage, groundwater recharge and streamflow were simulated for 59 mountain catchments in Czechia in the period 1980–2014 using a bucket-type catchment model. The model output was evaluated against observed daily runoff and snow water equivalent. Hypothetical scenarios were performed, which allowed to analyse the effect of inter-annual variations in snow storage on seasonal runoff separately from other components of the water balance.

The results showed that 17–42% (26% on average) of the total runoff in the study catchments originates as snowmelt, despite the fact that only 12–37% (20% on average) of the precipitation falls as snow. This means that snow is more effective in generating catchment runoff compared to liquid precipitation. This was demonstrated by modelling experiments which showed that total annual runoff and groundwater recharge decrease in the case of a precipitation shift from snow to rain. In general, snow-poor years were clearly characterized by a lower snowmelt runoff contribution compared to snow-rich years in the analysed period. Additionally, snowmelt started earlier in these snow-poor years and caused lower groundwater recharge. This also affected summer baseflow. For most of the catchments, the lowest summer baseflow was reached in years with both relatively low summer precipitation and snow storage. This showed that summer low flows (directly related to baseflow) in our study catchments are not only a function of low precipitation and high evapotranspiration, but they are significantly affected by the previous winter snowpack. This effect might intensify drought periods in the future when generally less snow is expected.

# 1    Introduction

Mountain catchments are often influenced by snow, which significantly affects seasonality in runoff. However, snow water equivalent (SWE) has been decreasing in many mountains regions over the last decades and spring snowmelt tends to occur earlier in the year (Beniston, 2012; Fyfe et al., 2017; Harpold et al., 2012; Klein et al., 2016; Marty et al., 2017b). This suggests that snow and snowmelt dynamics respond to increasing air temperature due to climate change (Barnett et al., 2005; Bavay et al., 2013; Jenicek et al., 2018; Marty et al., 2017a). The higher air temperature causes a shift from snowfall to rain resulting in

lower snowfall fraction, a proportion of snowfall water equivalent (hereafter referred to as snowfall) to annual precipitation. Consequently, the amount of snow and peak SWE are reduced as well (Berghuijs et al., 2014; Jenicek et al., 2016). As a response to increasing temperature, the snowmelt starts earlier in the year, which can result in slower snowmelt rates due to lower available energy, such as solar radiation (Klein et al., 2016; Musselman et al., 2017).

Reduced snow accumulation, and earlier and slower snowmelt cause earlier and lower groundwater recharge (Beaulieu et al.,

2012; Foster et al., 2016). For the groundwater recharge, the topography is important as the water is transported from steep terrain surrounding mountain ridges to lower elevations (Carroll et al., 2019). Therefore, higher elevations are important for catchment storage (Floriancic et al., 2018; Hood and Hayashi, 2015; Staudinger et al., 2017) as well as for stabilizing the streamflow at lower elevations especially during drought periods (Carroll et al., 2019; Cochand et al., 2019). Higher snowpack disproportionly feeds groundwater leading to more streamflow (Barnhart et al., 2016).

The decreasing snow storages and groundwater recharge in mountain regions further influence summer low flows (Dierauer et al., 2018; Ledvinka, 2015; Li et al., 2018; Van Loon et al., 2015; Potopová et al., 2016). Higher snowpack and later snowmelt contribute to summer baseflow and increase the period for which snowmelt contributes to streamflow (Hammond et al., 2018; Langhammer et al., 2015). For catchments at higher elevations, this period may cover the whole summer (Godsey et al., 2014; Jörg-Hess et al., 2014). Earlier snowmelt and melt-out in the future will further shorten this period (Etter et al., 2017; Jenicek

et al., 2018). For summer low flows in Europe, the liquid precipitation and evapotranspiration is usually more important than previous winter snow storages (Floriancic et al., 2019), but low snowpack causes a decrease in summer low flows in the case of simultaneously low summer precipitation (Jenicek et al., 2016).

Several earlier studies focused on identification of physical mechanisms that would help explain how earlier or later snowmelt influences the runoff generation (summarized in Barnhart et al., 2016). One mechanism describes that due to higher air

temperature during earlier snowmelt, a greater proportion of snowmelt evaporates (Barnhart et al., 2016; Bosson et al., 2012). In contrast, earlier snowmelt occurs in periods when vegetation is less active, thus using less water. Therefore, more water flows into the stream. Another effect is that slower snowmelt (associated with earlier snowmelt; Trujillo and Molotch, 2014) leads to lower streamflow generation due to the fact that during later and faster snowmelt, soil moisture might be more often above its capacity leading to higher streamflow. Additionally, higher snowmelt rates lead to higher baseflow (Barnhart et al.,

2016). Therefore, it is important to analyse inter-annual variability of snow, climate and streamflow characteristics in areas where snow is an important source for runoff generation.

The role of snow in seasonal catchment runoff, summer low flows and water supply is frequently quantified using several snow-related metrics, such as peak SWE or snowfall fraction (Curry and Zwiers, 2018; Hammond et al., 2018). When a modelling approach is applied, the ratio of snowmelt runoff to total runoff is often used (Jenicek et al., 2018; Li et al., 2017;

Stahl et al., 2016). Such approaches enable to quantify the importance of snow in the generation of spring and summer runoff and in the "memory effect" of a specific catchment, i.e., how long snow affects runoff after snowmelt (Godsey et al., 2014; Jenicek et al., 2016). Several studies also showed that snow is more effective in generating the runoff compared to liquid precipitation. For example, in the western United States 53% of the total runoff originates from snowmelt, despite the fact that only 37% of the precipitation falls as snow (Li et al., 2017). The mentioned discrepancy between snowfall amount and

snowmelt runoff might have substantial impacts on seasonal distribution of the runoff and water supply in a warming climate, when more rainfall is expected compared to snowfall (Harpold et al., 2017; Safeeq et al., 2016).

Snowfall fraction might be an interesting indicator to what degree snow affects summer low flows. For example, Meriö et al. (2019) found a threshold of snowfall fraction of 0.35 in their study catchments in Finland. In catchments with snowfall fraction above the threshold, the summer low flows were sensitive to inter-annual variations in snow storages. Below the threshold,

summer precipitation and air temperature together with catchment characteristics were more important. Although specific thresholds might differ across world regions, the role of snow storages on summer low flows was shown in several studies from Europe and North America (Godsey et al., 2014; Jenicek et al., 2016).

The above studies show that changes in snow storages and their impact on runoff and groundwater recharge are in the centre of the current research. However, there is still a need to investigate mutual interactions between individual characteristics, such

as snow storages in snow-poor and snow-rich years and their interaction with groundwater recharge, snowmelt runoff, spring and summer rainfall, summer baseflow and low flows. This is specifically important for assessment of how the changes in snow storages affect seasonal runoff distribution and whether or not they may also affect annual water balance. The need to explain this variability and change of temporal and spatial processes also arose from the hydrology community initiative to identify major unsolved scientific problems in hydrology (Blöschl et al., 2019).

Therefore, the objectives of this study were to quantify 1) how inter-annual variations in snow storages affect spring and summer runoff, including summer low flows and 2) the importance of the snowmelt in generating the runoff compared to rainfall at different elevations with different snowfall fractions. We quantified the influence of inter-annual variability in snow storages on summer runoff and low flows in 59 catchments in Czechia with significant snowmelt contribution to runoff. Focusing on non-alpine region of Central Europe is important, as most of the studies addressing this topic were conducted for

alpine regions characterized by higher elevations and partly different climate conditions. Therefore, the identification of other snow-dominated regions which might become more vulnerable to drought occurrence in the future is critical. For snowmelt contribution, we used an "effect tracking" algorithm which is now accepted and used by the modelling community aiming to track the effect of individual water sources in the system to assess either the inter-annual variability of runoff components or their potential changes in the future (Weiler et al., 2018).

## 2 Data and methods

### 2.1 Study area and input data

We selected 59 mountain catchments with minor influence of human activity together with significant snow influence on runoff (Fig. 1, Table 1). An important criterion for the selection was the availability of long-term time series of hydrological and meteorological observations (~35 years). With our selection, we covered most of the mountain regions of Czechia, namely the Bohemian Forest (BF, 12 catchments), Ore Mts. (OM, 2 catchments), Western Sudetes (WS, 16 catchments), Central Sudetes (CS, 5 catchments), Eastern Sudetes (ES, 13 catchments) and Western Carpathians (WC, 11 catchments). Although the mean catchment elevation ranges only from 491 to 1297 m a.s.l, all catchments have the seasonal snowpack every year. The mean $SWE_{max}$ for individual catchments ranges from 35 mm for the lowest catchments to 664 mm for the highest catchments (Table 1).

The daily runoff data were available for all closure profiles of the catchments from the Czech Hydrometeorological Institute (CHMI). Similarly, daily precipitation, daily mean air temperature and weekly SWE data from selected climatological stations located within or nearby individual catchments were obtained from the CHMI. Stational data were further used in a hydrological model. All data were available for the period from 1980 to 2014, except three catchments, from which data started in 1981 or 1982.

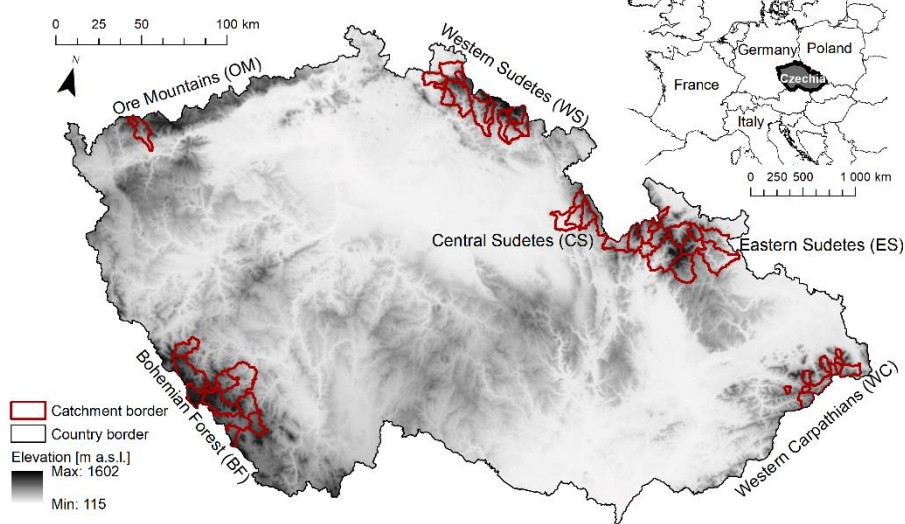

**Figure 1: Location of the study catchments in Czechia.**

**Table 1: Selected characteristics of the study catchments. Catchments IDs are given in the following way: 100 – Bohemian Forest (BF), 200 – Ore Mts. (OM), 300 – Western Sudetes (WS), 400 – Central Sudetes (CS), 500 – Eastern Sudetes (ES), 600 – Western Carpathians (WC). Snowfall fraction, $SWE_{max}$ and snowmelt contribution to runoff represent catchments means (1980–2014) resulting from model simulations.**

| ID | Name | Station | Area [km²] | Mean elevation [m a.s.l] | Elevation range [m a.s.l] | Mean slope [°] | Snowfall fraction [-] | $SWE_{max}$ [mm] | Snowmelt contribution to runoff [%] |
|---|---|---|---|---|---|---|---|---|---|
| 101 | Vydra | Modrava | 89.8 | 1140 | 983–1345 | 3.7 | 0.24 | 187 | 29 |
| 102 | Otava | Rejstejn | 333.6 | 1017 | 598–1345 | 5.1 | 0.19 | 113 | 22 |
| 103 | Hamersky | Antygl | 20.4 | 1098 | 978–1213 | 3.3 | 0.21 | 130 | 23 |
| 104 | Ostruzna | Kolinec | 92.0 | 755 | 541–1165 | 4.6 | 0.13 | 43 | 17 |
| 105 | Spulka | Bohumilice | 104.6 | 804 | 558–1131 | 4.9 | 0.13 | 35 | 17 |
| 106 | Volynka | Nemetice | 383.4 | 722 | 430–1302 | 4.7 | 0.13 | 38 | 17 |
| 107 | T. Vltava | Lenora | 176.0 | 1010 | 765–1314 | 5.2 | 0.17 | 83 | 22 |
| 108 | Blanice | Blanicky M. | 85.5 | 892 | 757–1197 | 3.5 | 0.16 | 45 | 21 |
| 109 | Blanice | Podedvor. M. | 202.8 | 844 | 558–1274 | 4.6 | 0.14 | 37 | 19 |
| 110 | Stassky | Novy Dvur | 9.9 | 962 | 792–1131 | 6.4 | 0.17 | 65 | 23 |
| 111 | T. Vltava | Chlum | 347.3 | 939 | 733–1314 | 4.9 | 0.15 | 66 | 19 |
| 112 | S. Vltava | Cerny Kriz | 102.4 | 921 | 738–1353 | 4.3 | 0.15 | 69 | 20 |
| 201 | Rolava | Chaloupky | 18.7 | 902 | 826–956 | 2.2 | 0.26 | 229 | 31 |
| 202 | Rolava | Stara Role | 125.3 | 761 | 398–994 | 4.3 | 0.19 | 112 | 24 |
| 301 | Jerice | Chrastava | 76.0 | 493 | 295–862 | 4.7 | 0.12 | 50 | 17 |
| 302 | C. Nisa | Straz | 18.3 | 672 | 368–850 | 4.6 | 0.19 | 121 | 24 |
| 303 | L. Nisa | Prosec | 53.8 | 611 | 419–835 | 4.4 | 0.17 | 94 | 20 |
| 304 | Smeda | Bily p. | 26.5 | 817 | 412–1090 | 8.3 | 0.29 | 236 | 35 |
| 305 | Smeda | Frydlant | 132.7 | 588 | 297–1113 | 5.9 | 0.18 | 112 | 24 |
| 306 | Jizera | D. Sytová | 321.8 | 771 | 399–1404 | 6.4 | 0.29 | 241 | 36 |
| 307 | Mumlava | Janov | 51.3 | 970 | 625–1404 | 7.8 | 0.33 | 383 | 41 |
| 308 | Jizerka | D. Stepanice | 44.2 | 842 | 490–1379 | 8.6 | 0.25 | 203 | 32 |
| 310 | M. Labe | Prosecne | 72.8 | 731 | 376–1378 | 6.3 | 0.21 | 131 | 27 |
| 311 | Cista | Hostinne | 77.4 | 594 | 358–1322 | 5.1 | 0.16 | 84 | 21 |
| 312 | Modry | Modry dul | 2.6 | 1297 | 1076–1489 | 13.0 | 0.38 | 664 | 41 |
| 313 | Upa | H. Marsov | 82.0 | 1030 | 581–1495 | 9.7 | 0.33 | 334 | 41 |
| 314 | Upa | H. S. Mesto | 144.8 | 902 | 452–1495 | 8.6 | 0.29 | 245 | 35 |
| 315 | C. Nisa | Uhlirska | 1.8 | 816 | 786–850 | 2.2 | 0.28 | 257 | 34 |
| 316 | C. Desna | Jezdecka | 4.8 | 899 | 792–1007 | 5.6 | 0.33 | 396 | 38 |
| 317 | Kamenice | Bohunovsko | 178.8 | 699 | 345–1069 | 5.8 | 0.23 | 171 | 28 |
| 401 | Bela | Castolovice | 214.1 | 491 | 269–1104 | 3.3 | 0.12 | 59 | 20 |
| 402 | Knezna | Rychnov/Kn. | 75.4 | 502 | 305–861 | 3.2 | 0.12 | 54 | 18 |
| 403 | Zdobnice | Slatina/Zd. | 84.1 | 721 | 395–1092 | 5.2 | 0.22 | 171 | 30 |
| 404 | D. Orlice | Klasterec/Orl. | 153.6 | 728 | 505–1078 | 4.8 | 0.19 | 151 | 25 |
| 405 | T. Orlice | Sobkovice | 98.5 | 622 | 459–965 | 4.6 | 0.18 | 91 | 25 |
| 501 | Branna | Jindrichov | 90.3 | 794 | 474–1378 | 6.8 | 0.16 | 117 | 21 |

| 502 | Desna | Sumperk | 246.9 | 736 | 320–1454 | 8.2 | 0.17 | 89 | 23 |
|-----|-------|---------|-------|-----|----------|-----|------|-----|-----|
| 503 | Moravice | Velka Stahle | 168.6 | 800 | 549–1415 | 5.4 | 0.23 | 108 | 32 |
| 504 | Opava | Krnov | 369.2 | 668 | 315–1437 | 6.0 | 0.12 | 53 | 18 |
| 505 | Opavice | Krnov | 173.3 | 547 | 318–912 | 5.1 | 0.17 | 50 | 25 |
| 506 | Morava | Vlaske | 96.5 | 790 | 448–1374 | 7.9 | 0.22 | 160 | 29 |
| 507 | Krupa | Habartice | 109.3 | 756 | 480–1267 | 6.6 | 0.26 | 188 | 34 |
| 508 | Telcsky | Stare mesto | 21.9 | 802 | 548–1102 | 6.4 | 0.16 | 132 | 20 |
| 509 | Morava | Raskov | 350.0 | 745 | 380–1378 | 6.9 | 0.25 | 165 | 33 |
| 510 | Bela | Jesenik | 118.0 | 799 | 443–1390 | 8.4 | 0.22 | 124 | 27 |
| 511 | Stribrny | Zulova | 21.4 | 712 | 390–1108 | 7.7 | 0.14 | 91 | 17 |
| 512 | C. Opava | Mnichov | 51.0 | 814 | 579–1186 | 6.6 | 0.16 | 88 | 18 |
| 513 | Opava | Karlovice | 150.9 | 854 | 503–1437 | 8.0 | 0.18 | 99 | 21 |
| 601 | V. Becva | V. Karlovice | 68.3 | 749 | 524–1042 | 6.8 | 0.17 | 107 | 22 |
| 602 | R. Becva | H. Becva | 14.1 | 745 | 568–966 | 7.0 | 0.21 | 130 | 27 |
| 603 | Celadenka | Celadna | 31.0 | 803 | 536–1187 | 9.9 | 0.21 | 124 | 27 |
| 604 | Ostravice | S. Hamry | 73.3 | 707 | 542–922 | 5.9 | 0.21 | 128 | 27 |
| 605 | Moravka | Uspolka | 22.2 | 763 | 560–1104 | 8.1 | 0.23 | 142 | 28 |
| 606 | Skalka | Uspolka | 18.9 | 785 | 571–1029 | 8.1 | 0.24 | 148 | 29 |
| 607 | Lomna | Jablunkov | 69.9 | 667 | 390–1011 | 7.5 | 0.19 | 96 | 23 |
| 608 | Mohelnice | Raskovice | 35.4 | 765 | 473–1209 | 10.3 | 0.22 | 131 | 27 |
| 609 | Slavic | Slavic | 15.1 | 827 | 575–1016 | 10.4 | 0.23 | 148 | 28 |
| 610 | Ropicanka | Reka | 12.2 | 696 | 454–1008 | 10.9 | 0.21 | 118 | 26 |
| 611 | Lesti | Solanec | 10.4 | 700 | 513–874 | 6.7 | 0.17 | 104 | 22 |

## 2.2 HBV model

To assess the impact of inter-annual variability of snow on streamflow, we need to simulate individual components of the rainfall-runoff process at a catchment level. For this, we used a bucket-type HBV model (Lindström et al., 1997) in its software implementation HBV-light (Seibert and Vis, 2012). Details on the model structure and routines are described in several studies
(Jenicek et al., 2018; Seibert and Vis, 2012).

The study catchments were sub-divided into elevation zones by 100 m reflecting the changes of precipitation and temperature with elevation. The time series of daily precipitation, daily mean air temperature and monthly potential evapotranspiration ($P_{ET}$) represent the main model inputs. The temperature-based method described by Oudin et al. (2005) was used for $P_{ET}$ calculation.

The HBV model was calibrated for each catchment against observed runoff and SWE using a genetic calibration algorithm (Seibert, 2000). The integrated multi-variable model calibration approach was applied using a combination of three goodness-of-fit criteria (Table 2); 1) model efficiency for runoff using logarithmic values ($R_{runoff}$) (Nash and Sutcliffe, 1970), 2) model efficiency for SWE ($R_{SWE}$) and 3) volume error ($R_{vol}$). The resulting objective function ($R_{weighted}$) was calculated from these

three criteria as a weighted average with *a*, *b* and *c* representing the weights for each criterion. Different weights were tested based on our experience with the model to produce as accurate as possible simulations considering both high and low flows, water balance and snow storages.

**Table 2: Objective functions used for model calibration and validation.**

| Objective function | Equation | Weights |
|---|---|---|
| Model efficiency for runoff | $R_{\text{runoff}} = 1 - \dfrac{\sum(\ln Q_{\text{obs}} - \ln Q_{\text{sim}})^2}{\sum(\ln Q_{\text{obs}} - \overline{\ln Q_{\text{obs}}})^2}$ | 60% |
| Model efficiency for SWE | $R_{\text{SWE}} = 1 - \dfrac{\sum(SWE_{\text{obs}} - SWE_{\text{sim}})^2}{\sum(SWE_{\text{obs}} - \overline{SWE_{\text{obs}}})^2}$ | 20% |
| Volume error | $R_{\text{vol}} = 1 - \dfrac{|\sum(Q_{\text{obs}} - Q_{\text{sim}})|}{\sum(Q_{\text{obs}})}$ | 20% |
| Weighted efficiency | $R_{\text{weighted}} = a \cdot R_{\text{runoff}} + b \cdot R_{\text{SWE}} + c \cdot R_{\text{vol}}$ | n.a. |

The observed time series were divided in two sub-series; first (1980-1997) was used for model calibration, second (1998-2014) for model validation. The two periods covered both cold and warm years and wet and dry years. Although mean annual air temperature increased by 0.7°C for the validation period, annual precipitation and peak SWE did not differ significantly between both periods (mean $SWE_{\text{max}}$ was 141 mm for the calibration period and 140 mm for the validation period; annual precipitation was 1104 mm for the calibration period and 1143 mm for the validation period).

The parameter uncertainty was addressed by performing 100 calibration runs resulting in 100 parameter sets. These 100 sets were further used to create 100 simulations. A median simulation was used for further analyses. This median simulation was derived in a way, that individual daily values for specific simulated variables (runoff, SWE, groundwater recharge etc.) were calculated as a median from all 100 respective values resulting from simulations. This procedure for the model set-up and calibration was also used in Jenicek et al. (2018), although for a different region.

## 2.3    Modelling experiments

To study the effect of inter-annual variations in snow storages on seasonal runoff characteristics (such as baseflow, deficit volumes and recharge) separately from other meteorological controls (mainly liquid precipitation and actual evapotranspiration, $A_{\text{ET}}$), the hypothetical scenarios were modelled. These simulations consist in changing the threshold temperature $T_{\text{T}}$ (a parameter included in the HBV snow routine) which is used to differentiates between snow and rain. By changing the $T_{\text{T}}$, we can control the amount of accumulated snow and snowmelt timing, while other variables remain unaffected (such as total amount of precipitation). Therefore, the $T_{\text{T}}$ was progressively changed from -5°C to +5°C. Changes in this parameter influenced the simulated snowfall and thus SWE, snowmelt onset, melt rates and melt-out. The experiment was applied to the whole study period and thus capturing a variety of meteorological conditions. In this experiment, the snowfall correction factor ($S_{\text{FCF}}$) correcting solid precipitation for the undercatch, was set to "1" (meaning no correction applied to

original input precipitation data). Similarly, $P_{ET}$ was not adjusted according to the inter-annual variations in air temperature

(meaning only input daily $P_{ET}$ was used, calculated from long term data as described earlier). This enabled a separation of the effect of changing $T_T$ on snow characteristics and seasonal runoff from other potential effects. With this procedure, we were able to attribute simulated runoff changes to changes in winter conditions.

## 2.4    Snow and streamflow signatures

We calculated several snow, groundwater and streamflow characteristics to analyse the impact of inter-annual variations in

snow accumulations on seasonal runoff. These characteristics were calculated from median simulations as described in Section 2.2.

Snow conditions for individual years of the study period were represented by February to May maximum SWE ($SWE_{max}$) which represents the late winter snow maximum. The snowfall fraction ($S_f$) describes the phase of precipitation. The snowfall fraction was calculated as the fraction of annual snowfall to annual precipitation using a single threshold temperature $T_T$. The

165 values of $T_T$ for individual catchments resulted from the model calibration and ranged from -1.58°C to 1.13°C.

The snow component of the streamflow ($Q_s$) was simulated by the HBV model. The method to track both rain and snow components, so called "effect tracking", is based on complete mixing of the two components in a "virtual mixing tank" (Stahl et al., 2016; Weiler et al., 2018). There are two assumptions for calculation; 1) the liquid water occurring in the snowpack is either considered as snow (in the case of melt of the snowpack) or rain (in the case of rain on snow), 2) when refreezing of the

170 liquid water in the snowpack is simulated, the source is considered as snow. The two assumptions mean that the source of the input water (precipitation) can be changed only from rain to snow by refreezing. However, this process is rather negligible in absolute terms. Both daily and seasonal snow runoff ($Q_s$) and the fraction of snow runoff to annual runoff ($Q_{sf}$) were used in further analyses.

The groundwater recharge ($G_w$) was simulated by the HBV model. It represents the outflow from the soil box into groundwater

boxes defined by the model (Seibert and Vis, 2012). From simulated time series, the fraction of winter (Dec-Feb, $G_{w-DJF}$) and spring (March-May, $G_{w-MAM}$) recharges to total annual recharge were calculated.

While the winter and spring recharge is a useful indicator to show how snow contributes to groundwater storage, the summer baseflow is related to the state of groundwater storage and thus represents both summer precipitation inputs and previous (spring) precipitation and snowmelt groundwater recharge. The summer (June to August, JJA) baseflow ($Q_b$) was calculated

by the HBV model as an outflow from the lower groundwater box ($S_{LZ}$) which is a part of the model's response routine. The inflow into the $S_{LZ}$ is controlled by percolation (parameter $P_{ERC}$, mm d$^{-1}$) and outflow is controlled by a recession coefficient $K_2$ [d$^{-1}$] (Seibert and Vis, 2012).

The deficit volume ($D_v$) represents a water volume lacking in rivers below the defined threshold (Van Loon, 2015). Therefore, it is used as a measure to describe hydrological drought conditions. We used 90$^{th}$ percentile of the flow duration curve. We

also tested the 75$^{th}$ percentile without any major impact on the results. We used a variable level threshold method which uses different thresholds calculated separately for individual months in summer (June to August).

To show the inter-annual and seasonal differences in snow runoff in study catchments, most of the signatures were calculated for relatively snow-rich years and relatively snow-poor years. The snow-rich years were defined as years with annual $SWE_{max}$ above the third quartile of the study period (1980-2014), while the snow-poor years represent years with annual $SWE_{max}$ below the first quartile of the study period.

## 3 Results

### 3.1 Model calibration and validation

The results arising from model testing showed the overall good performance of the model both for the calibration and validation periods (Fig. 2). The model accurately simulated both runoff ($R_{runoff}$, $R_{vol}$) and SWE ($R_{SWE}$). The SWE was reproduced better at higher elevation catchments which contributed to mostly high agreement of observed and simulated runoff in terms of water balance and runoff seasonality. The calibration period shows slightly higher values of individual objective functions compared to those from the validation period. The results for the validation period represent more realistic model performance since calibration values represent rather the theoretically best possible model performance at the specific catchment.

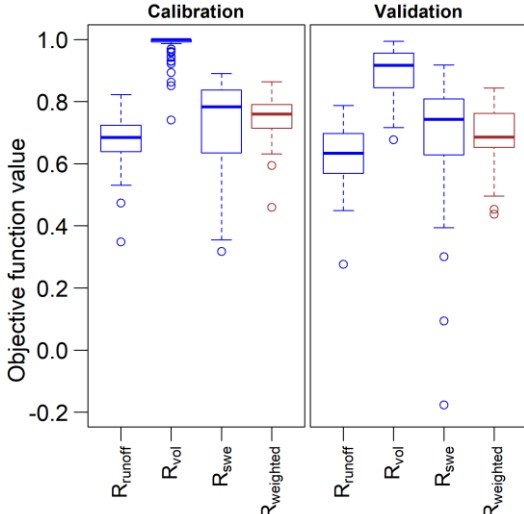

**Figure 2: Results of model calibration and validation for study catchments. The objective function value for each catchment represents a median from 100 parameter sets. Boxes represent 25th and 75th percentile (with median as a thick line), whiskers represent 1.5 multiplier of interquartile range.**

### 3.2 Relative importance of snow to runoff during snow-poor and snow-rich years

The snowmelt is more effective in generating the runoff compared to liquid precipitation (Fig. 3). On average, 26% of the annual runoff originated as snowmelt in our study catchments (17–42% for individual catchments for the study period), despite the fact that only 20% (12–37%) of the annual precipitation occurred as snow (Fig. 3a). Additionally, catchment runoff coefficients (the ratio between catchment precipitation and runoff) were higher for annual snowfall to snowmelt runoff than

for annual rainfall to rainfall runoff (results not shown). The higher runoff fraction for snow-generated runoff is caused mainly by lower actual evapotranspiration during winter. Both $S_f$ and $Q_{sf}$ are lower for snow-poor years (brown points) compared to

snow-rich years (blue points). It also seems that the difference between $Q_{sf}$ and $S_f$ was higher for snow-rich years (up to 12% for some catchments) and increased with elevation for catchments where the snow is relatively more important in generating the runoff (Fig. 3b, $p$ value $< 0.001$). These results might have important implications for annual runoff volume in the future when the decrease in snowfall fraction is expected.

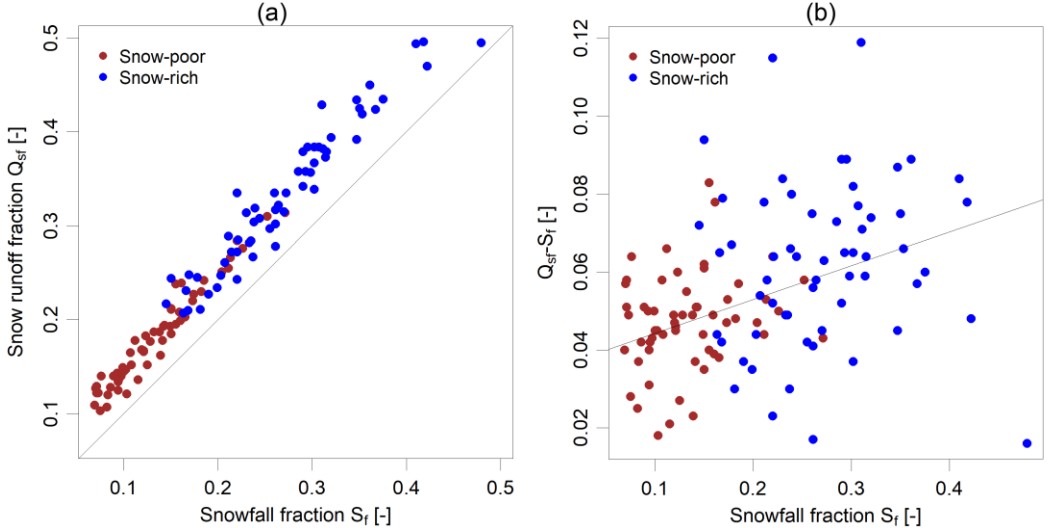

**Figure 3: (a) A relation between snowfall fraction ($S_f$) and snow runoff fraction ($Q_{sf}$) for all study catchments. (b) A dependence of $Q_{sf}$ and $S_f$ difference on snowfall fraction. Individual points represent mean snowfall fractions and snow runoff fractions for snow-poor years (brown points) and snow-rich years (blue points) for individual catchments.**

To show the seasonal distribution of snow runoff in study catchments, we calculated the relative contribution of snow runoff on total runoff ($Q_{sf}$) as an average for each day of the year. Then we compared this relative contribution of snow runoff for

relatively snow-poor years and snow-rich years (Fig. 4a). The figure shows a significantly lower snowmelt contribution for snow-poor years (up to 40%) in all catchments. The largest decrease in $Q_{sf}$ occurred at the end of April/beginning of May in the highest elevation catchments. This largest decrease in $Q_{sf}$ is somewhat shifted towards March in lower elevation catchments indicating that the snow runoff in lower elevation catchments occurs earlier in the year due to higher air temperatures and thus earlier snowmelt onset.

In contrast, there was an increase in relative snow contribution to runoff during snow-poor years in winter, which indicates that snow poor years were usually warmer compared to snow-rich years and thus the runoff increased due to more snowmelt periods during winter (Fig. 4a). Additionally, the decrease in relative snow runoff contribution also occurred during summer (June-August) which indicates that snow melted earlier in snow-poor years and less contributed to spring groundwater recharge and thus summer runoff.

While Fig. 4a shows the snow contribution to total runoff ($Q_{sf}$), Fig. 4b shows monthly differences in simulated total runoff for snow-poor and snow-rich years for four selected catchments  (Vydra, C. Nisa, Desna and Ostravice, see Table 1)

representing different geographical regions (four different regions), geology (granite rock, metamorphic rock or flysch) and elevations. As expected, March to May (sometimes even June) runoff was much lower for snow-poor years compared to snow-rich years due to lower snow storages and thus snowmelt runoff. In contrast, winter runoff increased in snow-poor winters due to more rain than snowfall (lower $S_f$) and thus winter runoff occurred without delay. Summer runoff differed little between snow-rich and snow-poor years. Interestingly, total annual runoff in snow-poor years was often much lower compared to snow-rich years. This may be partly explained by higher effectiveness of snowfall to generate the runoff (Fig. 3) or by the fact that less snow in specific years was connected not only to warmer air temperatures, but also to the lack of precipitation (results not shown). Therefore, this lack of winter precipitation caused the decrease in total annual runoff depth in our study catchments during snow-poor years.

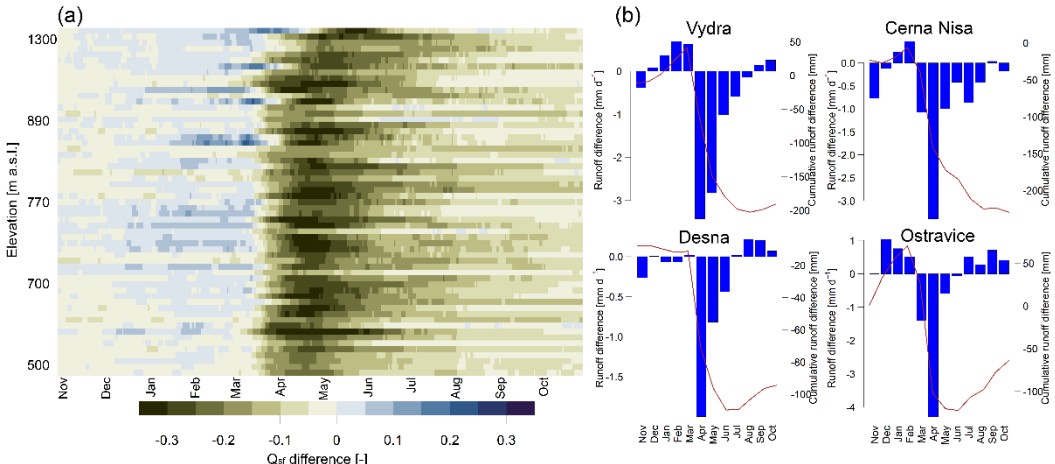

**Figure 4: (a) Difference between mean daily $Q_{sf}$ [-] for snow-poor and snow-rich years. Rows represent individual catchments sorted from top to bottom according to mean catchment elevation from highest to lowest (y-axis not-to-scale), columns represent day of year. (b) Difference between mean monthly runoff (blue bars) and cumulative monthly differences in runoff (red line) for snow-poor and snow-rich years; (b1) Vydra (Bohemian Forest), (b2) Cerna Nisa (Western Sudetes), (b3) Desna (Eastern Sudetes), (b4) Ostravice (Western Carpathians).**

Lower snow storages in snow-poor years compared to snow-rich years led to lower snowmelt runoff contribution (Fig. 5a) and thus lower seasonal groundwater recharge. This seasonal recharge was expressed as a fraction of December to May recharge on total annual recharge (Fig. 5b). These recharge fractions are clearly lower for the entire cold season for snow-poor years in 57 out of 59 catchments (97%), with higher differences for catchments with higher $S_f$. However, recharge fractions were higher for the period from December to February (DJF) and lower from March to May (MAM) for snow-poor years (not shown) indicating that winter liquid precipitation during snow-poor years led to higher recharge during winter and thus partly compensated the lower recharge fraction during spring. The lower seasonal recharge fraction also caused lower annual recharge for snow-poor years compared to snow-rich years despite the fact that annual precipitation was almost the same for both groups

(results not shown). Lower annual groundwater recharges probably caused lower annual runoff in snow-poor years compared to snow-rich years (Fig. 5c). This partial result supports the results shown in Fig. 3.

Lower snow storages, snowmelt runoff and spring recharge also caused lower summer baseflow in 42 out of 59 catchments (71%), although only for 26 catchments was the difference higher than 5 mm (a sum for JJA period), affecting partly catchments with higher $S_f$ (Fig. 5d). Additionally, summer baseflow strongly negatively correlates with summer deficit volumes (median Spearman rank correlation for all catchments reached the value of -0.8). It resulted in higher summer deficit volumes in snow-poor years compared to snow-rich years for 47 out of 59 catchments (80%, Fig. 5e). Additionally, the difference in deficit volumes between snow-poor and snow-rich years was negatively correlated (*p* value < 0.05) with difference in snowmelt runoff between snow-poor and snow-rich years with usually larger absolute difference for catchments with higher $S_f$ (Fig. 5f).

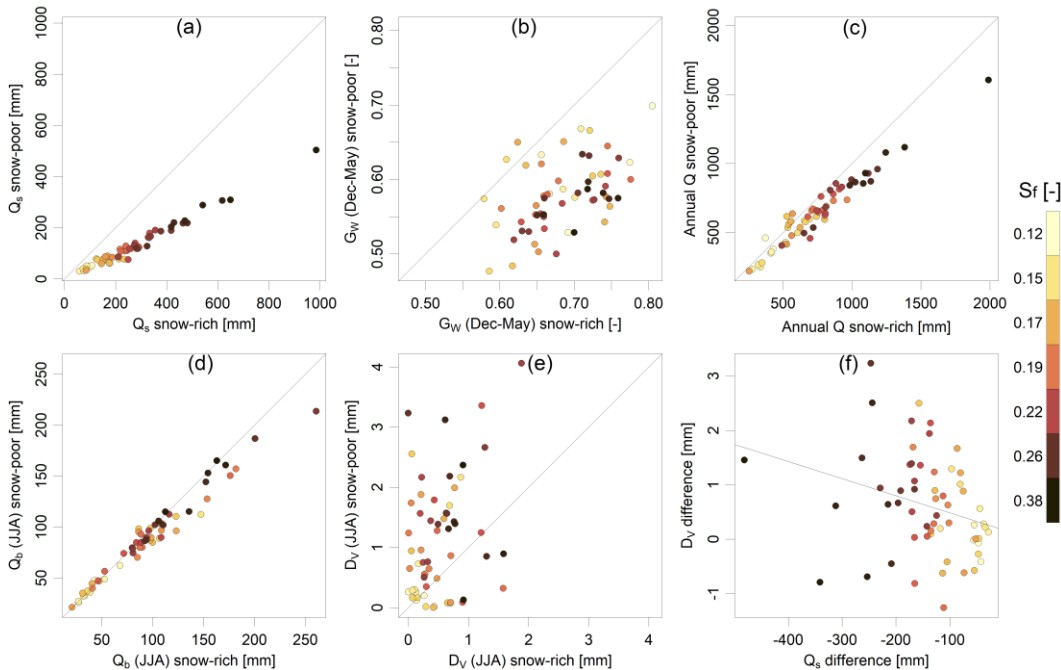

**Figure 5: Difference in selected signatures for snow-rich and snow-poor years for study catchments; (a) Annual snowmelt runoff $Q_s$, (b) Seasonal recharge fractions $G_W$ (Dec–May), (c) Annual runoff $Q$, (d) Summer baseflow $Q_b$ (JJA), (e) Deficit volumes $D_V$ (JJA), (f) Relation of deficit volumes $D_V$ (JJA) difference to snow runoff $Q_s$ difference between snow-poor and snow-rich years. Colour scale used for snowfall fraction.**

Figure 4 and Fig. 5 showed that less snow led to lower snowmelt contribution to total runoff and to lower winter and spring groundwater recharge. Besides, there was lower baseflow and higher deficit volumes in most catchments in snow-poor years. However, the figures do not provide us with information about how important snow storages are in influencing summer baseflow and deficit volumes compared to summer precipitation. Therefore, we analysed the relation between relative anomalies in summer (JJA) precipitation and relative anomalies in $SWE_{max}$ compared to summer baseflow (Fig. 6). For this

analysis, the same four catchments, as previously shown in Fig. 4, were selected to demonstrate that summer baseflow is associated with both summer precipitation and annual $SWE_{max}$ (Fig. 6). The lowest summer baseflow is associated with both the lowest summer precipitation and $SWE_{max}$ (dark brown points are mostly located in the bottom left quadrants in individual panels of Fig. 6). Although, summer precipitation seems to be more important for baseflow amount, Fig. 6 indicates that for some unit precipitation amount, the baseflow was lower for years with low annual $SWE_{max}$. Nevertheless, the described behaviour differs for individual catchments. For example, the summer precipitation seems to be crucial for summer baseflow in the Ostravice catchment (Fig. 6d), while snow does not play an important role in influencing summer baseflow (the darkest brown points are located roughly equally in both bottom-left and bottom-right quadrants).

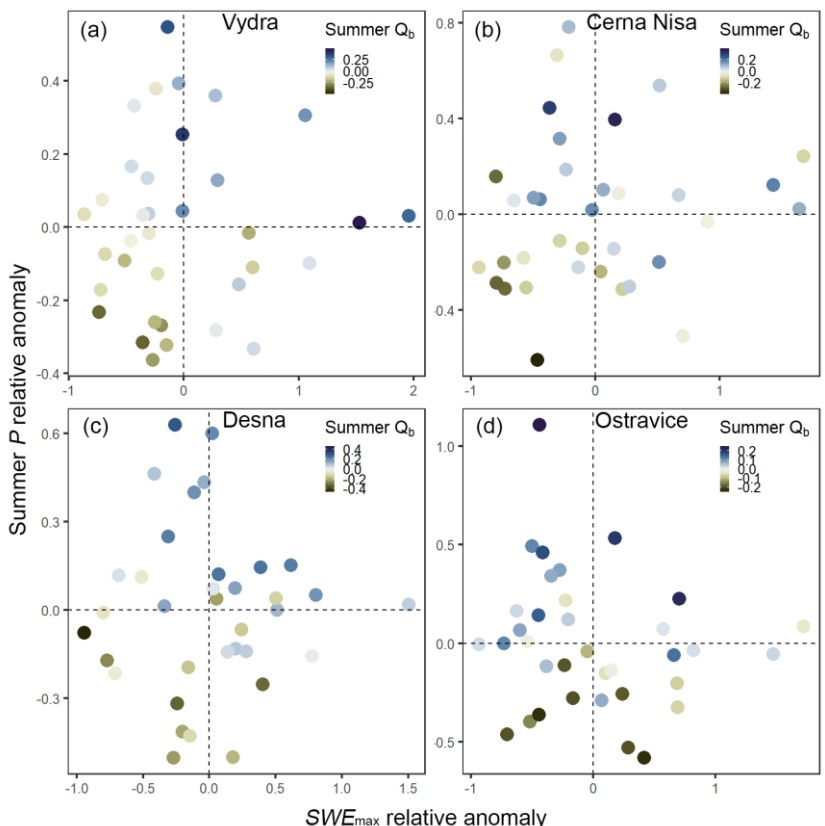

**Figure 6: Dependence of summer baseflow ($Q_b$) relative anomalies on $SWE_{max}$ and summer (JJA) precipitation relative anomalies at four selected catchments. (a) Vydra (Bohemian Forest), (b) Cerna Nisa (Western Sudetes), (c) Desna (Eastern Sudetes), (d) Ostravice (Western Carpathians)**

Similar mutual relationships between $SWE_{max}$, summer precipitation and summer baseflow, as shown in Fig. 6, were explored and generalized for all catchments (Fig. 7) to present the relative importance of annual $SWE_{max}$ and summer precipitation to summer baseflow. Figure 7a depicts the median summer baseflow relative anomalies for years with below-average summer precipitation (x-axis) against the median summer baseflow relative anomalies for years with below-average $SWE_{max}$ (y-axis).

The Figure 7 revealed that 1) below-average summer baseflow occurred for 58 out of 59 catchments (98%) for below-average summer precipitation (points located to the left of the $x=0$ line) and for 40 out of 59 catchments (68%) for below-average

$SWE_{max}$ (points located below the $y=0$ line) and 2) below-average summer precipitation generated lower baseflow than below-average $SWE_{max}$ (points located above the one-to-one line). This implies that snow is important in generating baseflow in our study catchments, but summer precipitation is more important. This is specifically valid for catchments in the Eastern Sudetes (ES) and Western Carpathians (WC) where summer precipitation seems to be the dominant driver for summer baseflow generation (the darkest points in Fig. 7a are located above the $y=0$ line and thus show positive baseflow anomalies for below-

average $SWE_{max}$).

Figure 7b shows that baseflow is lowest for both below-average precipitation and below-average $SWE_{max}$. However, for 41 out of 59 catchments (69%), the baseflow for years with below-average precipitation increased when there was simultaneously above-average $SWE_{max}$ (points are located above the one-to-one line). Moreover, 13 out of 59 catchments (22%) showed even positive summer baseflow anomalies for above-average $SWE_{max}$ despite the negative anomaly of the summer precipitation

(points located above the $y=0$ line). For those catchments (mostly smaller catchments with a greater proportion of area at higher elevations), snow storages seem to be more important for summer baseflow than summer precipitation.

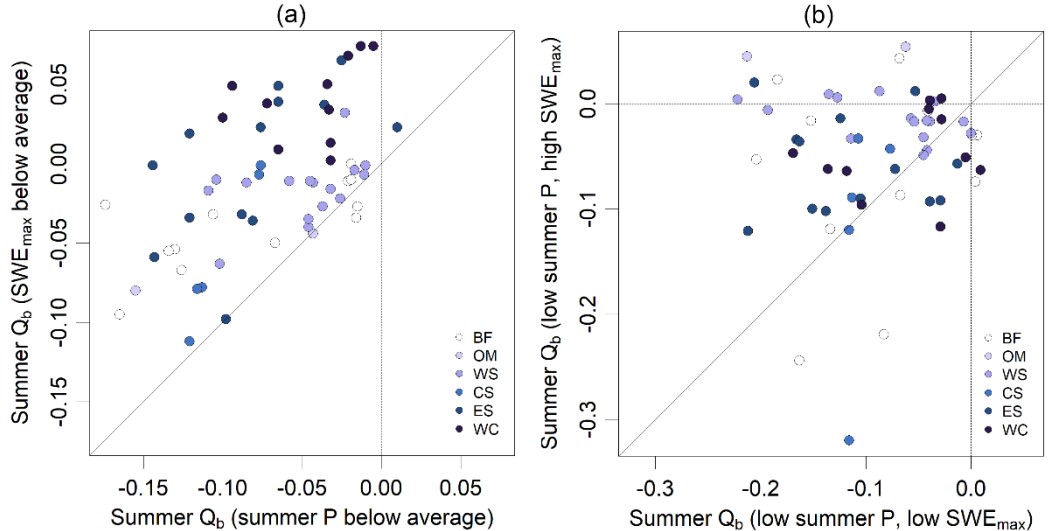

**Figure 7: (a) Relation of median summer baseflow for years with below-average summer precipitation and years with below-average**
**$SWE_{max}$. (b) Median summer baseflow for years with both below-average summer precipitation and below-average $SWE_{max}$ related to median summer baseflow for years with both below-average summer precipitation and above-average $SWE_{max}$. Points represent individual catchments, colour represents region of the specific catchment (see Section 2.1 for region abbreviations).**

### 3.3    Modelling changes in runoff for snowfall-rain transition

The results presented in Section 3.1 indicated that less snow caused a decrease in spring recharge and thus a decrease in
summer baseflow and an increase in summer deficit volumes. However, the above approach did not allow to fully split the

effect of snow storages on summer runoff from the effects of other meteorological drivers, mainly summer precipitation and $A_{ET}$. Therefore, we performed a simple modelling experiment simulating the transition of snowfall to rain while the total annual precipitation and air temperature remained unchanged (see methods).

The model simulated lower snowfall and thus decrease in snow storages and shorter snow-covered season as a response to threshold temperature $T_T$ increase (result not shown). This snow storages decrease caused a decrease in both annual and summer runoff for all 59 catchments (Fig. 8a, Fig. 8b). Simulated snow decrease resulted in lower groundwater recharge in winter and spring for 56 out of 59 catchments (95%, Fig. 8c). Expectedly, the groundwater recharge increased in winter months (Dec-Feb) thanks to earlier snowmelt and lower $S_f$ (and thus rain infiltrated into the soil immediately). However, the decrease in groundwater recharge in spring (March-May) was much larger than the increase in recharge in winter (not shown), resulting in an overall decrease in Dec-May recharge.

While the decrease in both summer and annual runoff provides the information about overall water availability regardless of the seasonal distribution and extreme situations, the summer deficit volumes provide information about water availability during most critical situations, such as summer low flows. The modelling experiment showed an increasing trend in deficit volumes with decreasing $S_f$ for most of the catchments (Fig. 8d). In contrast to the decrease in annual runoff with decreasing $S_f$, which was simulated for all study catchments, the increase in summer deficit volumes was simulated only for 34 out of 59 catchments (58%). The remaining catchments do not show clear tendency of deficit volumes or they behave in the opposite direction, which indicates more complex behaviour of such catchments (e.g. due to specific catchment properties, such as geology or slope steepness). However, when looking at June to August deficit volumes separately, the deficit volumes increased for 80% of the catchments in June, for 52% in July, and for 55% in August (results not shown). This evolution suggested the decreasing role of snow in influencing deficit volumes in the analysed months. Similar results as for summer deficit volumes were also achieved for the number of days with deficit volumes, which also mostly increased when the snowfall fraction decreased (results not shown).

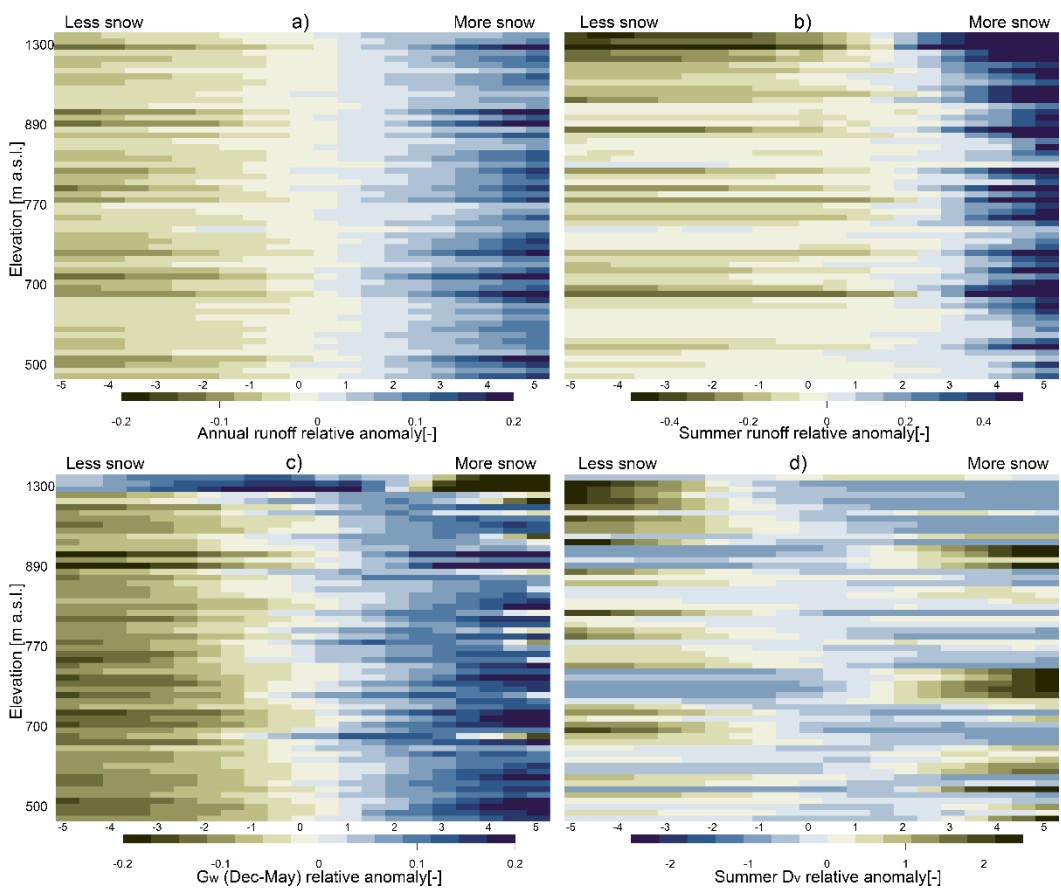

**Figure 8: Relative change of selected signatures with increasing $T_T$ for study catchments. (a) annual runoff, (b) summer (JJA) runoff, (c) groundwater recharge (Dec-May) and (d) summer (JJA) deficit volumes. Rows represent individual catchments sorted from top to bottom according to mean catchment elevation from highest to lowest (y-axis not-to-scale), columns represent $T_T$ used in model simulations. Colours show normalized values relative to their means (different scales used for individual panels).**

## 4 Discussion

### 4.1 HBV model set-up and parameters uncertainty

Since the presented results are based on HBV model runoff simulations, the uncertainty arising from the model parametrization needs to be addressed. This was done by 100 model calibration trials resulting in 100 parameter sets. This way the model generated more robust results. Additionally, a multi-variable approach was used for calibration to correctly simulate both SWE and runoff. This procedure led to better model performance especially in higher elevation catchments with higher snowpack as also shown in other studies (Etter et al., 2017; Jenicek et al., 2018; Seibert, 2000).

The interpretation of our results partly relies on the ability of the model to simulate groundwater storage. Although the model allows to use the groundwater level for calibration, next to discharge and SWE, we did not use the groundwater observations to calibrate the model. The reason was that the density of the measuring network does not allow to find the groundwater

stations (either boreholes or springs as a proxy) which would sufficiently represent the whole catchment since the spatial variability in groundwater storages in a catchment is very large due to the variability in geology and soils. In contrast, the streamflow used to calibrate the model represents the integrated output from the whole catchment, and also the observed SWE data usually represents the catchment snow storage well enough (at least at a specific elevation zone). Additionally, we were concentrated on relative differences (year-to-year variations) between groundwater fluxes in individual catchments rather than on absolute values. Some studies also showed that catchment storage calculated using different methods (water balance calculations, recession curve analysis, HBV modelling) are, in general, comparable and correlated, although the quantitative estimates may differ (Staudinger et al., 2017).

There are several issues related to model parametrization. Many of the model parameters might have an important effect on result interpretation. For example, $P_{ET}$ calculated for a specific day did not account for actual air temperature on that day, but it only reflected long-term mean of daily air temperature for the full study period (1980–2014). This affected simulated $P_{ET}$ values which then did not reflect inter-annual variability due to inter-annual variability of air temperature. Nevertheless, by neglecting the potential feedbacks from $P_{ET}$ inter-annual variability, this approach enabled a better separation of snow influence on summer low flows.

Modelling experiments also opened further questions related to model structure and parameterization, specifically how individual model procedures and parameters represent the real natural processes. For example, a snowfall correction factor ($S_{FCF}$) is used in the model to correct solid precipitation for the undercatch (e.g. due to wind). Nevertheless, it may also compensate some processes not explicitly included in the model, such as snow interception, sublimation and $A_{ET}$ from snow cover. Similarly, threshold temperature $T_T$ used in our modelling experiment to control snowfall fraction is also used to set the snowmelt onset and as a threshold temperature to distinguish whether the $S_{FCF}$ will be applied or not. Therefore, $S_{FCF}$ was set to "1" in the modelling experiment to avoid modification of the solid precipitation amount throughout the experiment and thus entire water balance. For the future model development, it might by therefore useful to set one $T_T$ for snow/rain separation and another $T_T$ for snowfall correction. Similarly, the calculation of $A_{ET}$ during existing snow cover on the ground might be beneficial.

For studies performed in snow-dominated catchments, it is also important how well (or badly) the model simulates snow storages at different elevations. For example, Girons Lopez et al. (2020) tested several modifications of the HBV model snow routine in Swiss and Czech catchments (a subset of those used in this study) and showed that the snow routine employed in the HBV model provided relatively good results, although some modifications might be worth consideration, such as using seasonally-variable melt factor or exponential snowmelt function. Nevertheless, an increased model complexity does not necessarily mean the better model ability to reproduce SWE and runoff (Girons Lopez et al., 2020). The above-mentioned parameter issues might be important when using HBV or similar bucket-type models for impact studies, such as modelling the impact of climate change on catchment runoff.

## 4.2    Influence of snow storages on snowmelt runoff and groundwater recharge

The results showed that the fraction of runoff originating as snowmelt is by 2–12% higher than the fraction of precipitation occurring as snow (higher values for snow-rich years and for catchments at higher elevation). This indicates that snowmelt is more effective in generating runoff compared to liquid precipitation in the study catchments. The reason for such behaviour is lower $A_{ET}$ during winter and lower water demand by vegetation (the latter is not included in the HBV model structure), both resulting in lower precipitation losses. Besides, the $A_{ET}$ is calculated only for a snow-free ground in the HBV model. Another reason might result from the fact that soil moisture is more often at its field capacity, thus the model also simulates higher runoff during snowmelt events (Barnhart et al., 2016; Li et al., 2017). In other words, snowmelt rates control the relative partitioning of snowmelt water between evapotranspiration and streamflow (Barnhart et al., 2016).

Nevertheless, in absolute values, snowfall represented a higher water volume than runoff from snow which comes out from the definition of the "effect tracking" algorithm in HBV (as described in Section 2.4). However, in few catchments it happened that few hydrological years showed higher snowmelt runoff than snowfall in that year since part of this snowmelt contribution came from the previous year (probably thanks to a long catchment storage). The mentioned effect is certainly worth investigating further, but it goes beyond the scope of the current study.

The effect of a higher snowmelt runoff fraction than snowfall fraction was also shown by Li et al. (2017) in the western United States using a similar modelling approach and it is also supported by the results achieved by Berghuijs et al. (2014) who showed that higher snowfall fractions generate higher annual runoff in the western United States. Although our results are limited to the study catchments and may not be easily generalized, the described catchment behaviour might have an important impact on runoff generation in the future, where the shift from snow to rain during winter is predicted due to the increase in air temperature (Jenicek et al., 2018).

The results showed that the contribution of snow to runoff is much lower for snow-poor than snow-rich years in the study catchments. The decrease was obvious also for the period from June to August, indicating that snow also contributed to summer runoff for snow-rich years. The snow contribution to summer runoff, including low flows, was also documented by several other studies (Godsey et al., 2014; Jenicek et al., 2016; Stahl et al., 2016), although mostly for higher elevation catchments with later snowmelt compared to our study catchments. The summer snow runoff originates from spring snowmelt, which propagated into deeper groundwater layers and contributed to runoff with delay. The snow runoff contribution was calculated using the "effect tracking" algorithm implemented in the HBV model using "virtual mixing tanks" with limited capacities. The method was tested within several studies (Stahl et al., 2016; Weiler et al., 2018). The algorithm is a useful approach to assess changes in discharge components (Weiler et al., 2018).

The average contribution of groundwater to streamflow was from 22% to 84%, with higher values for catchments with generally higher snowfall fractions and thus higher snow storages. However, besides snow amounts, some other regional differences were found. For example, catchments in the Bohemian Forest (rather flat catchments on metamorphic or granite rock with a large portion of peatland) had larger groundwater contributions than catchments in the Western Carpathians (more

steep catchments on flysch). Nevertheless, the potential influence of basin attributes on catchment storage and runoff generation needs to be further investigated. The relative fraction of groundwater in streams is sensitive to inter-annual variation in snow storages, as also shown by Carroll et al. (2019). Similarly, the dynamic groundwater storage maybe correlated with elevation, indicating the relation between the groundwater storages and snow storages (Staudinger et al., 2017).

### 4.3 Influence of snow storages on summer baseflow and deficit volumes

Inter-annual variations in snow storages also affected summer baseflow, which is, additionally to spring snowmelt, related to summer precipitation and evapotranspiration. Our results showed that less snow led to lower snowmelt contribution to total runoff and to lower spring groundwater recharge with larger differences for catchments with higher $S_f$. These results correspond to other studies (Carroll et al., 2019; Cochand et al., 2019; Meriö et al., 2019). However, it does not necessarily mean that summer baseflow and potentially low flows are lower as well, since both the baseflow and low flows are more influenced by other water balance components, such as precipitation and evapotranspiration during spring and summer (Floriancic et al., 2019; Jenicek et al., 2016). Since the baseflow is a major runoff component during low-flow periods, it can be used as an indicator showing the potential extremity of such periods. Our results showed that snow-rich years produced higher summer baseflow in our study catchments. This indicated that spring snowmelt increased groundwater storage. Although the lower baseflow does not mean that potential low-flow periods are more likely to occur, it indicates that if the low-flow period occurs (e.g. due to lack of summer precipitation and/or high $A_{ET}$), the minimum streamflow might drop to lower values.

The results shown in Fig. 5 indicated that larger catchments with larger elevation ranges have more complex behaviour, especially in case of summer baseflow and deficit volumes. The impact of snowfall fraction (and thus snow storages) on summer baseflow and deficit volumes is less obvious or not present in these catchments, most likely due to the effect of higher $A_{ET}$ in lower parts caused by higher air temperatures. Therefore, the snow is less important for summer runoff and low flows in such catchments.

The above effect of large area and elevation range could also explain the fact that the relationship between snow storages and summer runoff may be better explained by snowfall fraction rather than elevation. Although the snowfall fraction significantly increases with mean catchment elevation in our study catchments (Pearson correlation coefficient is 0.53, $p$ value<0.001), the correlation coefficient might be influenced by the fact that mean catchment elevation cannot describe the hypsography of individual catchments.

Although in most of the catchments the summer precipitation was more important for summer baseflow, in some catchments with the highest snowfall fractions and larger proportion of area located at higher elevations, the winter snowpack was probably of a similar importance to the summer precipitation. An important implication for the future climate is that the summer baseflow might be lower because of lower snow storages even when summer precipitation would not change. The snow as the dominant mechanism controlling summer low flows was also proved by Meriö et al. (2019) for Finnish catchments with snowfall fractions higher than 0.35. Similar to the mentioned study, we also discovered some regional differences between our study catchments. For example, for catchments in the Eastern Sudetes and Western Carpathians, summer precipitation

dominated the summer baseflow despite relatively high snow storages. This might indicate that those catchments had a shorter "memory effect" of snow to influence the runoff. Thus, the water exchange is faster in these catchments resulting in shorter residence times. Our results did not provide possible reasons for such behaviour, but differences in climate regimes (increasing continentality from west to east), geology (flysch in the Western Carpathians vs. metamorphic or granite rock in other regions) and morphology (higher slopes in the Western Carpathians) could provide some explanation, as shown by several authors for other regions (Floriancic et al., 2018; Li et al., 2018; Staudinger and Seibert, 2014). However, more detailed research would be necessary.

Nevertheless, understanding how snow storages in snow-poor and snow-rich years influence summer baseflow and deficit volumes is always limited due to different meteorological conditions in individual years. For example, winter seasons in years with lower snow storages were also warmer and dryer, which could affect summer baseflow and low flows. However, despite these differences in winter precipitation and temperatures, summer precipitation, summer air temperature and summer $A_{ET}$ were almost the same for both snow-poor and snow-rich years. This suggests that differences in summer runoff signatures can be related to changes in snow storages and spring groundwater recharge.

## 4.4    Snowfall-rain transition and potential future impacts

The results related to catchment response in snow-rich and snow-poor years indicated important potential consequences for annual and seasonal runoff and deficit volumes, which might decrease (or increase, for deficit volumes) in the future when the decrease in snowfall fraction is expected. However, the hypothesis highlighting the importance of snowfall fraction for runoff amount cannot be fully proven by splitting the study period into snow-poor and snow-rich years, due to the fact that snow-poor and snow-rich years differed not only in snow storages, but also in other meteorological signatures (as explained in Section 4.3). Therefore, this runoff volume decrease was proven by a modelling experiment simulating the progressive change from snowfall to rain leaving the total precipitation, air temperature and $P_{ET}$ unchanged. This way, we were able to separate the effect of changing snow to rain from other water balance components. A similar approach, using the same model and parameters (but for different purposes), was applied also by Jenicek et al. (2018).

The changes in runoff due to snowfall-rain transition as simulated by modelling experiments pointed at two different aspects; 1) changes in annual water balance and 2) changes in seasonal runoff distribution. The first aspect is demonstrated by results of the hypothetical experiment which showed an increase in annual runoff for all 59 catchments. A closer look at the results suggests that the model simulated lower snowfall and thus snow storages that melted earlier. Due to more days without snow cover, the total annual $A_{ET}$ increased ($A_{ET}$ is calculated for days without snow cover in the model) which caused a decrease in total annual runoff. It means that the increasing contribution of liquid precipitation to total runoff cannot compensate for the lower contribution of solid precipitation to total runoff. Moreover, potential snow-rain transition in the future will be caused by an increase in air temperature, which was not changed in our modelling experiment. Therefore, the $A_{ET}$ might be even higher due to the temperature increase (in the case of enough available water). Consequently, annual runoff will likely decrease even more than indicated by our experiment. However, we are aware that this particular result may be affected by the model structure

which describes the whole rainfall-runoff process in a simplified way, and thus the real catchment behaviour might not be captured correctly.

The second aspect, changes in seasonal runoff distribution, was caused mainly by lower snow accumulation for lower snowfall fractions (more rain than snowfall) and by earlier snowmelt. This widely influenced the timing of groundwater recharge and thus spring and summer streamflow, low flows and deficit volumes (Fig. 8b-d). In contrast to the decreasing annual runoff due to decreasing snowfall fraction, the increase of summer deficit volumes was simulated only for 58% of the study catchments. The remaining catchments did not show a clear trend or they behaved in the opposite direction. This suggested more complex behaviour of these catchments, which is probably caused by their location rather at lower elevations, and thus lower snow storages. Therefore, other climatic variables and catchment properties, such as geology and related groundwater storages might be more important than snow storages. However, more research would be necessary to find a detailed explanation. Nevertheless, the results from modelling experiments are consistent with analyses based on snow-poor and snow-rich years. Similar results to those for summer deficit volumes were achieved also for the number of days with deficit volumes, which also increased when snowfall fraction decreased. The described changes in seasonal runoff distribution are, in our opinion, more important (although expected) since they widely determine the water availability during the warm period when the water demand is generally higher (for vegetation growth, agriculture, hydropower production etc.).

Although our results may not be easily generalized since they are limited to the specific region, the decreasing annual runoff in the case of a precipitation shift from snow to rain suggests that this shift in the precipitation phase will change the catchment behaviour such that less water might be available for summer runoff in the future. The lower annual runoff might be critical for water supply and water reservoir management (Brunner et al., 2019). For the seasonal water balance, it will therefore be important to understand whether the future increase in winter precipitation predicted by climate models for the region of Central Europe can compensate for the expected future reduction in the snowmelt component.

## 5    Conclusions

We found that 17–42% (26% on average) of the total runoff in our selected study catchments originates as snowmelt, despite the fact that only 12–37% (20% on average) of the precipitation falls as snow (Fig. 3). It also seems that the difference is increasing at higher elevations with higher relative importance of snow for runoff regime. This particular result suggests that snow is more effective in generating catchment runoff compared to liquid precipitation. This might have an important impact on water availability in the case of a future decrease in snow.

The mentioned difference between snowfall fraction and snowmelt runoff fraction was also documented by modelling experiments which showed that total annual runoff decreased in the case of a precipitation shift from snow to rain, even in the case where the total amount of precipitation and $P_{ET}$ remained unchanged (Fig. 8). This might imply lower annual catchment runoff in the future when a precipitation shift from snow to rain due to air temperature increase is predicted by climate models.

In general, snow-poor years were clearly characterized by lower snow runoff contribution to total runoff compared to snow-rich years in the analysed period 1980-2014 (Fig. 5). Additionally, snowmelt started earlier in these snow-poor years and influenced the runoff for a shorter period compared to snow-rich years. Snow-poor years generated lower annual groundwater recharge and annual runoff compared to snow-rich years despite similar annual precipitation and $A_{ET}$, which resulted in mostly higher deficit volumes.

Inter-annual variations in snow storages also affected summer baseflow, which is, besides snow, related to summer precipitation and evapotranspiration. For most of the catchments, the lowest summer baseflow was reached in years with both relatively low summer precipitation and snow storage (Fig. 7). This showed that summer low flows (directly related to baseflow) are not only the function of low summer precipitation, but they are significantly affected by the previous winter snowpack. Although the summer precipitation is usually the most important climatic factor controlling the summer low flow, the decrease in snow and earlier snowmelt might intensify the future summer low flows in mountain catchments when generally less snow is expected.

Modelling experiments indicated that the future decrease in snowfall fraction together with changes in seasonal runoff distribution might result in lower annual runoff despite no changes in total precipitation (Fig. 8). The lower annual runoff might be critical for water supply and water reservoir management. For the seasonal water balance, it will therefore be important whether the future increase in winter precipitation predicted by climate models for the region of Central Europe can compensate for the expected future reduction in the snowmelt component.

# 6    Data availability

Meteorological and hydrological data for the calibration of the HBV model were obtained from the Czech Hydrometeorological Institute (contact person: Ondrej Ledvinka, ondrej.ledvinka@chmi.cz). The HBV model outputs are available from the first author upon request.

# 7    Author contribution

MJ initiated the study, developed the methodology and performed all analyses. OL prepared input meteorological and hydrological data used to calibrate the HBV model. MJ prepared the manuscript with a contribution from OL.

# 8    Competing interests

The authors declare that they have no conflict of interest.

## 9 Acknowledgements

Support from the Czech Science Foundation (project no. 18-06217Y, Influence of seasonal snowpack on summer low flows: climate change implications on hydrological drought) is gratefully acknowledged. Meteorological and hydrological data for the calibration of the HBV model were obtained from the Czech Hydrometeorological Institute (contact person: Ondrej Ledvinka, ondrej.ledvinka@chmi.cz). Many thanks are due to Tracy Ewen for English corrections.

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
