# Peer review of "Importance of snowmelt contribution to seasonal runoff and summer low flows in Czechia"

_Hydrology and Earth System Sciences, 2019_

## Referee Comment (RC1) · Anonymous Referee #1 · 27 Jan 2020

This paper investigates the role of snow (and rain) on streamflow across 59 Czech catchments. The objectives of the study are: to quantify how snow storages affect spring and summer runoff and to quantify how much runoff snowmelt in generates compared to rainfall. The study uses data of 50 catchments and simulations using the HBV model. They show the following results: 1. Snow runoff fractions exceed snowfall fractions (Fig 3), from which they conclude snow produces more runoff than rain. 2. How much runoff occurs in particular months varies between snow rich and snow poor years (Fig 4), with overall more runoff in snow rich years. 3. Several streamflow signatures vary between snow rich and snow poor years (Fig 5). 4. Summer base flow depends on both SWE and summer P (Fig 6+7) 5. That also in models annual and

summer runoff strongly depend on the snow fraction (Fig 8).

These results are generally useful for the HESS' readership, as they address the important issue of how snow (and its anticipated future changes) affect river flow. However, before I can recommend publication of this article I think several things need to be addressed first:

- This HBV results suggest that snow produces runoff differently than rain. However, HBV treats snowmelt and rainfall largely similarly. This seems counterintuitive (or a paradox). It needs to become clearer in the modeling results how snowmelt is different than rain that leads to these runoff differences. Otherwise, I am not sure what we really learn from the presented results.

- The results listed above as 1-4 have all be shown before or are mostly trivial. There might be value in showing this again for the study catchments, but then I think the paper should better explain what we learn about the hydrology of these places, rather than largely use them as data for making some general statements.

- All results of groundwater recharge rely on the model output of an unvalidated flux (since no GW data are used). How do we have confidence they reflect actual groundwater recharge behaviours?

- The paper contains a lot of unclear statements or language that if (interpreted as written) is wrong. I made a list of suggestions below, but this list is far from comprehensive. Please check the paper another time critically. This is really important because for too many statements it remains unclear what the authors claim to be true, and thereby makes it even impossible to review

Detailed comments

L9: add the word "often" (or something similar), otherwise this general statement is false.

L11: (and in winter runoff). Not necessary to state, but maybe not bad to mention.

L14: model output, not model performance.

L15: the simulations are not "hypothetical" as they have been performed. I the paper intends to say something like "Hypothetical scenarios were modelled"

L19-20: "This was documented by [. . .] from snow to rain" This does not seem to be a logical statement. Maybe change the verb "documented"?

L22: would "reduced" be more specific than "affected" and therefore more informative?

L29: "largely affects" seems a bit odd. Maybe "often affects" or "can affect".

L30: "tend to occur" not "occurs".

L32: "to increase [. . .] climate changes". Why mention "precipitation"? And reword "to increase in air temperature" to "to increasing air temperatures". (And probably make "climate changes" singular).

L34: "during winter" may be an unnecessary (and sometimes wrong) specification here. In many mountain areas the shift from snow towards rain will be biggest in spring and fall (when temperatures are often near 0) compared to winter (when temperatures are generally below zero even when it gets a bit warmer)

L34: "a rate of"? I do not think this makes sense here. Please check what is intended to be said here.

L35: would "reduced" be more specific than "affected" and therefore more informative?

L37-38: I understand why you say "On the contrary" but this only makes sense by having the reader guess that this has an opposite effect on total streamflow generated (which you don't say, nor make it clear that this is what you're thinking about). Therefore I would try to reword this a little.

L39-40: "Changes in [. . .] and occurs earlier". Or "Reduced snow accumulation, and earlier and slower snowmelt cause earlier and less groundwater recharge (Beaulieu et

al., 2012; Foster et al., 2016)."

L41: to "lower elevations" (make plural)

L44-45: "Higher snowpack generates higher groundwater flow driven by snowmelt rates and thus contributes more to streamflow 45 (Barnhart et al., 2016)." Does not seem to be a logical statement. Do you mean something like "Higher snowpack disproportionally feed groundwater leading to more to streamflow (Barnhart et al., 2016)"?

L54: "were" seems redundant.

L57: Thus "using" not "uses"

L96: Consider removing "the" at the start of the sentence.

L110-111: "For this, we used a bucket-type HBV model (Lindström et al., 1997) in its implementation, called HBV-light (Seibert and Vis, 2012)" This sentence is clear, but I would recommend to rephrase it. (E.g. remove 'in its implementation")

L121: "Different weights were tested to achieve the best possible performance of the model" This seems somewhat vague and arbitrary. What made you choose the particular weight in the end (i.e. what made them the "best")?

L133-134: "The similar procedures for model set-up and calibration was also used earlier in (Jenicek et al., 2018), although in different region." Fix the language of this sentence. For example by: something like "This procedure for model set-up and calibration was also used in Jenicek et al. (2018), although for different region".

L137: the simulations are not "hypothetical" as they have been performed. I the paper intends to say something like "Hypothetical scenarios were modelled"

L182: I am unsure what "simulated correctly" would really mean here. Do you mean "accurately simulated"?

L192-200: It is unclear to me to what extent these results originate from snow being

more effective in producing runoff than rain or whether this is because of the seasonal timing of precipitation (independent whether it's snow or rain).

L435: "This particular result proves that snow is more effective in generating catchment runoff compared to liquid precipitation" seems like an overly strong statement. Tone down the word "prove" and choose something like "indicates" or "suggests".

L454-456: ". An understanding of potential model artifacts might be important ..." is very vague. Can it be made more specific?
* * *

---

## Author Comment (AC1) · 3 Feb 2020

**Authors' response to interactive comment of the anonymous Referee #1**

Black text: Referee comment

Blue text: Authors' response

We thank the reviewer for the valuable comments and suggestions to improve our contribution. We provide point-by-point reply below.

This paper investigates the role of snow (and rain) on streamflow across 59 Czech catchments. The objectives of the study are: to quantify how snow storages affect spring and summer runoff and to quantify how much runoff snowmelt in generates compared to rainfall. The study uses data of 50 catchments and simulations using the HBV model. They show the following results: 1. Snow runoff fractions exceed snowfall fractions (Fig 3), from which they conclude snow produces more runoff than rain. 2. How much runoff occurs in particular months varies between snow rich and snow poor years (Fig 4), with overall more runoff in snow rich years. 3. Several streamflow signatures vary between snow rich and snow poor years (Fig 5). 4. Summer base flow depends on both SWE and summer P (Fig 6+7) 5. That also in models annual and summer runoff strongly depend on the snow fraction (Fig 8).

These results are generally useful for the HESS' readership, as they address the important issue of how snow (and its anticipated future changes) affect river flow. However, before I can recommend publication of this article I think several things need to be addressed first:

- This HBV results suggest that snow produces runoff differently than rain. However, HBV treats snowmelt and rainfall largely similarly. This seems counterintuitive (or a paradox). It needs to become clearer in the modeling results how snowmelt is different than rain that leads to these runoff differences. Otherwise, I am not sure what we really learn from the presented results.

Thank you for this valuable comment. The changes in runoff due to snowfall/rain transition as simulated by modelling experiments (Fig. 8) pointed at two different aspects; 1) changes in annual water balance and 2) changes in seasonal runoff distribution. The first aspect was shown in Fig 8a and, in model, was caused by lower actual evapotranspiration (AET) for higher snowfall fractions due to more days with snow cover (AET is calculated only for days with no snow cover on the ground in the model). This is mainly discussed on lines 410-418 (section 4.4). We are aware that this particular result is influenced by the model structure which describes the whole rainfall-runoff process in a simplified way and thus the real catchment behaviour might not be captured correctly.

The second aspect, changes in seasonal distribution, was caused mainly by lower snow accumulation for lower snowfall fractions (more rain than snowfall) and by earlier snowmelt. This widely influenced the timing of groundwater recharge and thus spring and summer streamflow, low flows and deficit volumes (Fig. 8b-d). This second aspect is, in our opinion, more important (although expected) since it widely influences the water availability during the warm period when the water need is generally higher (for vegetation growth, agriculture, hydropower etc.).

We will describe it in a clearer way in the revised version (in results and discussion sections).

- The results listed above as 1-4 have all be shown before or are mostly trivial. There might be value in showing this again for the study catchments, but then I think the paper should better explain what we learn about the hydrology of these places, rather than largely use them as data for making some general statements.

We agree that most of findings are not surprising as they mostly support our existing qualitative knowledge of how snow contributes to spring and summer runoff. However, we believe that the findings are still important even if they do not change our process understanding, and the quantification is a valuable and novel contribution. Besides, we were concentrated on non-alpine (outside the Alps) region of Central Europe where there is only a little published information on how ongoing changes in snow storages and snow/rain distribution at different elevations affect seasonal distribution of runoff. This is specifically important for identification of regions which might become more vulnerable to drought occurrence in the future. We also benefit from modelling approach enabling us to simulate both snow and rain runoff components and thus track the snow and rain signal in runoff. We believe that the fact that our focus was on the interplay of different rainfall-runoff components goes beyond what has been done before and thus it might bring some new insight into this topic.

Nevertheless, we are aware that our results are limited to the specific region and may not be easily generalized. Therefore, it was our intention to write the text in this respect. However, maybe this is not always clear from our formulations. Therefore, we will go through the text again to describe it clearer, to better highlight the novelty and to put more emphasis on regional consequences of our results.

- All results of groundwater recharge rely on the model output of an unvalidated flux (since no GW data are used). How do we have confidence they reflect actual groundwater recharge behaviours?

It is true that our results are not based on direct measurements of all individual runoff components since such measurements were not available for all analysed components. Therefore, we calibrated a model to simulate those components of the water cycle, for which observational data were not available. The HBV model, which was used in our study is, despite its simplicity (and thus limiting ability to represent rainfall-runoff process in a fully physical way), widely used and accepted by the scientific community, especially for impact modelling at a catchment scale. To better address the model uncertainty, we used an integrated multi-criteria approach to calibrate the model using three objective functions to validate our model against both streamflow and SWE. The model allows using also groundwater (GW) data for calibration, but these data are not easily available for all study catchments. Besides, the density of the measuring network does not allow us to find the GW stations (either boreholes or springs as a proxy) which would sufficiently represent the whole catchment since the spatial variability in GW storages in a catchment is very large due to the variability in geology and soils. In contrast, the streamflow used to calibrate the model represents the integrated output from the whole catchment, and also the observed SWE data usually represents the catchment snow storage well enough (at least at a specific elevation zone). Therefore, it is questionable to which degree the use of the GW data for model calibration would result in more accurate simulations.

Additionally, we were concentrated on relative differences (year-to-year variations) between groundwater fluxes in individual catchments rather than on absolute values. Some studies also showed (Staudinger et al. 2017) that catchment storage calculated using different methods (water balance calculations, recession curve analysis, HBV modelling) are, in general, comparable and correlated, although the quantitative estimates may differ. Therefore, we believe that using HBV simulations for assessing the relative inter-annual differences in GW storage is acceptable approach even when GW data were not used for model calibration.

We will include the above explanation into the discussion section to better describe all uncertainties and limitations of using such model for this topic.

- The paper contains a lot of unclear statements or language that if (interpreted as written) is wrong. I made a list of suggestions below, but this list is far from comprehensive. Please check the paper another time critically. This is really important because for too many statements it remains unclear what the authors claim to be true, and thereby makes it even impossible to review

Thank you for the comment. Although, the text was corrected by a native speaker (hydrologist), there could be still some unclear statements or wrong formulations. We will carefully go through the text again to correct all potential English errors.

**Detailed comments**

L9: add the word "often" (or something similar), otherwise this general statement is false.

Added.

L11: (and in winter runoff). Not necessary to state, but maybe not bad to mention.

Changed to "winter to summer runoff".

L14: model output, not model performance.

Changed.

L15: the simulations are not "hypothetical" as they have been performed. I the paper intends to say something like "Hypothetical scenarios were modelled"

Changed to "hypothetical scenarios".

L19-20: "This was documented by [: : :] from snow to rain" This does not seem to be a logical statement. Maybe change the verb "documented"?

"Documented" changed with "demonstrated",

L22: would "reduced" be more specific than "affected" and therefore more informative?

The word "affected" was used in the original manuscript because the baseflow was not reduced in all catchments (as explained in the next sentence). Therefore, we prefer to keep the sentence as is to avoid confusions, although it is less informative.

L29: "largely affects" seems a bit odd. Maybe "often affects" or "can affect".

Changed to "significantly" since the snow impact on runoff seasonality is really important and often substantial.

L30: "tend to occur" not "occurs".

Changed.

L32: "to increase [: : :] climate changes". Why mention "precipitation"? And reword "to increase in air temperature" to "to increasing air temperatures". (And probably make "climate changes" singular).

Changed.

L34: "during winter" may be an unnecessary (and sometimes wrong) specification here. In many mountain areas the shift from snow towards rain will be biggest in spring and fall (when temperatures are often near 0) compared to winter (when temperatures are generally below zero even when it gets a bit warmer)

We removed "during winter".

L34: "a rate of"? I do not think this makes sense here. Please check what is intended to be said here.

We changed "rate" with "proportion". The whole part is a definition of "snowfall fraction" (which is firstly used here) to avoid confusion about this term.

L35: would "reduced" be more specific than "affected" and therefore more informative?

Changed.

L37-38: I understand why you say "On the contrary" but this only makes sense by having the reader guess that this has an opposite effect on total streamflow generated (which you don't say, nor make it clear that this is what you're thinking about). Therefore I would try to reword this a little.

We removed the sentence since the mentioned information has no link to previous information and thus, we think it is redundant in this context.

L39-40: "Changes in [: : :] and occurs earlier". Or "Reduced snow accumulation, and earlier and slower snowmelt cause earlier and less groundwater recharge (Beaulieu et al., 2012; Foster et al., 2016)."

Changed, thank you for the suggestion.

L41: to "lower elevations" (make plural)

Changed.

L44-45: "Higher snowpack generates higher groundwater flow driven by snowmelt rates and thus contributes more to streamflow 45 (Barnhart et al., 2016)." Does not seem to be a logical statement. Do you mean something like "Higher snowpack disproportionly feed groundwater leading to more to streamflow (Barnhart et al., 2016)"?

Changed, thank you for the suggestion.

L54: "were" seems redundant.

Removed.

L57: Thus "using" not "uses"

Corrected.

L96: Consider removing "the" at the start of the sentence.

The sentence was reworded.

L110-111: "For this, we used a bucket-type HBV model (Lindström et al., 1997) in its implementation, called HBV-light (Seibert and Vis, 2012)" This sentence is clear, but I would recommend to rephrase it. (E.g. remove 'in its implementation')

We wanted to mention that we used HBV-light version of the model, which is the specific software implementation of the original HBV model. We slightly reworded the sentence.

L121: "Different weights were tested to achieve the best possible performance of the model" This seems somewhat vague and arbitrary. What made you choose the particular weight in the end (i.e. what made them the "best")?

Although we tested different weights, it is true that we did not use any consistent approach to find the best values of these weights. The testing was done just based on our experiences with the model and based on a literature. Therefore, it is true that the choice was rather arbitrary, although it reflected the main purpose of the model use (accurate simulation of both high and low flows, water balance and snow storages). We will reformulate the respective part to be clearer.

L133-134: "The similar procedures for model set-up and calibration was also used earlier in (Jenicek et al., 2018), although in different region." Fix the language of this sentence. For example by: something like "This procedure for model set-up and calibration was also used in Jenicek et al. (2018), although for different region".

Changed, thank you for the suggestion.

L137: the simulations are not "hypothetical" as they have been performed. I the paper intends to say something like "Hypothetical scenarios were modelled"

Changed.

L182: I am unsure what "simulated correctly" would really mean here. Do you mean "accurately simulated"?

Yes, we mean "accurately simulated". We changed it.

L192-200: It is unclear to me to what extent these results originate from snow being more effective in producing runoff than rain or whether this is because of the seasonal timing of precipitation (independent whether it's snow or rain).

We don't know whether we correctly understand this comment. The results here (Fig. 3) were analysed for individual hydrological years (1 Nov – 31 Oct) and thus any deviations from 1:1 line (Fig. 3a), in our opinion, should indicate the differences in annual water balance rather than differences in seasonal runoff distribution. We will add more explanation to the revised version.

L435: "This particular result proves that snow is more effective in generating catchment runoff compared to liquid precipitation" seems like an overly strong statement. Tone down the word "prove" and choose something like "indicates" or "suggests".

We agree, changed to "suggests".

L454-456: ". An understanding of potential model artifacts might be important : : :" is very vague. Can it be made more specific?

This conclusion refers to the issue about how the model structure could influence results discussed in section 4.1 (mainly L334-343). We agree that the mentioned formulation in the conclusion section could be more specific. We will reformulate the respective part of the text.

**References**

Staudinger, M., Stoelzle, M., Seeger, S., Seibert, J., Weiler, M. & Stahl, K. (2017) Catchment water storage variation with elevation. Hydrol. Process. 31(11), 2000–2015. doi:10.1002/hyp.11158.

---

## Referee Comment (RC2) · Anonymous Referee #2 · 22 Feb 2020

The paper presents how snow processes influence runoff generation in mountainous catchments in Czechia. The presented results are not novel, and similar things have been shown across different regions. However, the manuscript could still be a valuable contribution for the readership of HESS. The overall structure of the manuscript is quite clear, but inconsistent language makes the paper sometimes hard to follow, especially throughout the introduction and discussion. Below I suggest some changes that should be considered prior to publication.

At this point, I am not convinced by the conclusion that "snow is more effective in generating catchment runoff compared to liquid precipitation". First of all, it is not clear

what I actually see in Figure 3: Did you plot the mean of both groups (snow rich and snow poor) for every catchment? Please add some information to make this clearer. Second, I'd like to see the same calculations (Figure 3) with the absolute values for total snowmelt runoff and total snowfall precipitation as 26% (on average) of total runoff might still be less than 20% (on average) precipitation. Also the increasing trend with elevation in my opinion is not visible in the results. There needs to be further analysis (maybe cluster in elevation groups) to convince readers. I understand that some of these results are also supported by the HBV modelling. However, you need to more explicitly convince readers that snow vs. rainfall processes can be well separated in the current modelling setup.

A better characterization of the catchments (i.e., the runoff regimes, precipitation and runoff seasonality) is warranted. This will help to better emphasize why these results are valuable and why it might be useful to show the results for these specific study regions. To people who are not familiar with topography and hydroclimatology of Czechia it would be very helpful to have more "background" information on the study catchments. Please add a table with information on mean, max, min size, elevation, precipitation, temperature, discharge,. . . What are the main differences between the regions, and the four sample catchments? This is important to interpret the results afterwards (some of them are shown based on the different sample catchments). If I interpret the DEM correctly your highest peak is only 1602 m a.s.l., some of the catchments are far below 1000m in peak elevation, do they even have snowfall / accumulation every year? I find it difficult that, in the discussion section, you interpret the results based on the different regions, however they are not well characterized.

Detailed comments: line 98 you claim that the selection criterion is timeseries >35 years however in line 104 /105 you write that three catchments do have less data

line 125 although I tend to believe that annual precipitation, peak SWE did not change significantly it would be great to see this (maybe in a table in the supplementary)

[Figure]

line 155 what is the range of threshold temperature throughout the catchments?

Section 3.1 is not overly informative, in my opinion it can be moved to the supplement.

Figure 4 (and Figure 8): catchments are sorted by "mean" elevation, also add an arrow and write elevation next to y axis, and at least give starting and end value (115m a.s.l. to 1602m a.s.l.)

Figure 4 (and Figure 6): make it clear, that you show the results for four specific catchments maybe by using the catchment names as headlines for the subpanels)

Figure 5 and Figure 7: make sure that you use different color coding, as you show different things (in Figure 5 Sf and in Figure 5 the regions)

Figure 5 please mention the abbreviations (as in the axis titles) also in the figure caption

Figure 7 is a bit confusing: In panel (a), do you show a point for each catchment where x is the mean of baseflow from all years having below average summer precipitation and y is the mean of baseflow from all years having below average SWEmax? If that is what I see in Figure 7a, than 58 out of 59 catchments have below average summer baseflow when they experience below average summer precipitation. However, only 40 out of 59 catchments had lower summer baseflow when having lower SWE, which is not supporting your conclusion on the importance of SWE. Please revise this figure (and its caption) to make it clear what is shown.

In Figure 8 please consider using the same scale for the color bars to make the panels comparable.

Discussion: You mention data errors in the headline of 4.1 but you did not discuss them.

You need to better emphasize the challenges when separating liquid from solid precipitation within the HBV modelling framework. Maybe you can discuss the implications on your results a little more detailed.

The contribution from groundwater calculated with HBV is quite uncertain, you could also be looking at generally higher storage potential at higher elevations. Maybe you could consider discussing these uncertainties.

You mention a lot of interesting differences between the regions / catchments in the discussion, maybe you can add more information at an earlier part of the manuscript and build your story on these different regions.

Conclusions: I'd appreciate if you could relate the statements with the according figures, that makes it easier for the reader to recap on where to find the evidence for the conclusions

The second objective (lines 86 & 87) is to show the importance of snowmelt "at different elevations", however elevation differences where not really mentioned and I also did not find any concluding remarks regarding this statement.

I am also not convinced that I saw results that support that "future liquid precipitation will not compensate the lower solid precipitation", please re-write or leave out.

---

## Author Comment (AC2) · 8 Mar 2020

**Authors' response to interactive comment of the anonymous Referee #2**

Black text: Referee comment

Blue text: Authors' response

We thank the reviewer for the valuable comments and suggestions to improve our contribution. We provide point-by-point reply below.

The paper presents how snow processes influence runoff generation in mountainous catchments in Czechia. The presented results are not novel, and similar things have been shown across different regions. However, the manuscript could still be a valuable contribution for the readership of HESS. The overall structure of the manuscript is quite clear, but inconsistent language makes the paper sometimes hard to follow, especially throughout the introduction and discussion. Below I suggest some changes that should be considered prior to publication.

At this point, I am not convinced by the conclusion that "snow is more effective in generating catchment runoff compared to liquid precipitation". First of all, it is not clear what I actually see in Figure 3: Did you plot the mean of both groups (snow rich and snow poor) for every catchment? Please add some information to make this clearer.

Thank you for the comment. As you correctly assume, individual points in Figure 3 represent mean snowfall fractions and snow runoff fractions for snow-poor and snow-rich years for individual catchments. The snow-rich years were defined as years with annual $SWE_{max}$ above the third quartile of the study period (1980-2014), while the snow-poor years represent years with annual $SWE_{max}$ below the first quartile of the study period. Therefore, each point represents a mean calculated from 8-9 annual values derived from ~35-year-long time series. We will add more explanation to the figure caption to be clearer.

With Figure 3, we wanted to quantify to what degree the snowfall is important for runoff generation compared to rainfall in our study catchments. This was also assessed by runoff coefficients calculated separately for snowfall to snowmelt runoff and rain to rainfall runoff (values are not shown in the paper, but they are discussed in Section 3.2). The higher runoff fraction for snow-generated runoff is caused mainly by lower actual evapotranspiration during winter (see also our response to Referee 1). Additionally, in modelling experiments we showed that the transition of snowfall to rain caused changes in 1) annual water balance and 2) seasonal runoff distribution, affecting groundwater recharge and summer low flows (Fig. 8). We will add more explanation to the methods Section 2.4 (regarding snow runoff calculation in HBV), to results Section 3.2 (better description of Figure 3) and also to the related part of the discussion section (consequences for seasonal runoff distribution and water availability).

Second, I'd like to see the same calculations (Figure 3) with the absolute values for total snowmelt runoff and total snowfall precipitation as 26% (on average) of total runoff might still be less than 20% (on average) precipitation.

You are right that, in absolute values, snowfall represents a higher water volume than runoff from snow. By definition of the "effect tracking" algorithm in HBV (as described in Section 2.4), the source of the input water (precipitation) can be changed only by refreezing (from rain to snow), but this process is rather negligible in absolute terms. Therefore, snowfall will be always higher than snowmelt runoff over the defined longer period (losses from evapotranspiration will likely be always higher than changes of water source from rain to snow due to refreezing). Nevertheless, in few catchments it happened that few hydrological years showed higher snowmelt runoff than snowfall in

that year since part of this snowmelt contribution comes from previous year (probably thanks to long catchment storage). The mention effect is certainly worth for further investigation, but it goes beyond the scope of the current manuscript.

We carefully considered your suggestion to make a new figure with absolute snowfall and snowmelt runoff values, but we prefer not to include it since we think it would not provide any new information. Nevertheless, we will add more text regarding this issue to the methods (Section 2.4) and to the discussion to provide the reader with thorough explanation and interpretation.

Also the increasing trend with elevation in my opinion is not visible in the results. There needs to be further analysis (maybe cluster in elevation groups) to convince readers. I understand that some of these results are also supported by the HBV modelling. However, you need to more explicitly convince readers that snow vs. rainfall processes can be well separated in the current modelling setup.

You are right that we cannot make a direct conclusion regarding the effect of elevation. The increase in the difference between snowfall fraction and snow runoff fraction is statistically significant for catchments with higher snowfall fraction (see Fig. 3b) rather than elevation. In general, the elevation dependence is not directly evident from results, although some other results in our study (e.g. Fig. 5) indicated the relation with snowfall fraction, which generally increases with mean catchment elevation. However, the correlation of snowfall fraction and mean catchment elevation is not high, although significant (Pearson correlation coefficient is 0.53, p-value<0.001). The relatively lower correlation might be caused by the fact that mean catchment elevation cannot describe the hypsography of individual catchments and also by the fact that the variability of the mean catchment elevations is not high (800 m between the lowest and the highest catchment). We will clarify it better in the revised version of the manuscript (both results and conclusion sections). We will also add some more discussion on this topic.

A better characterization of the catchments (i.e., the runoff regimes, precipitation and runoff seasonality) is warranted. This will help to better emphasize why these results are valuable and why it might be useful to show the results for these specific study regions. To people who are not familiar with topography and hydroclimatology of Czechia it would be very helpful to have more "background" information on the study catchments. Please add a table with information on mean, max, min size, elevation, precipitation, temperature, discharge,: : :

Thank you for the suggestion. We will add a table showing the main catchment attributes and meteorological characteristics. We agree that this information might be useful for readers. Similar comment was also made by Referee 1. Additionally, we will put also more emphasis on regional differences of results as mentioned in one of your detailed comments bellow.

What are the main differences between the regions, and the four sample catchments? This is important to interpret the results afterwards (some of them are shown based on the different sample catchments). If I interpret the DEM correctly your highest peak is only 1602 m a.s.l., some of the catchments are far below 1000m in peak elevation, do they even have snowfall / accumulation every year? I find it difficult that, in the discussion section, you interpret the results based on the different regions, however they are not well characterized.

The four selected catchments represent different geographical regions and elevations. We will describe them in more detail in the revised version. For this, the new table with catchments characteristics (as mentioned in previous comment) might help as well.

Although the mean catchment elevation ranges only from 491 to 1297 m a.s.l, all catchments have the seasonal snowpack every year (mean $SWE_{max}$ for individual catchments ranges from 35 mm for lowest catchments to 664 mm for highest catchments). However, we agree that these regional differences are not well described both in results and discussion sections. We will add more details and interpretation to the revised version of the manuscript.

**Detailed comments**

line 98 you claim that the selection criterion is timeseries >35 years however in line 104 /105 you write that three catchments do have less data

It is correct that three catchments have shorter time series (by one or two years compared to the rest of catchments). We are aware that it may bring some inhomogeneity into results, but since the shortening is only one or two years, we decided to include those relatively snow-rich catchments to the analysis. We will describe it more clearly in the revised version of the manuscript.

line 125 although I tend to believe that annual precipitation, peak SWE did not change significantly it would be great to see this (maybe in a table in the supplementary)

Thank you for the comment. The mean $SWE_{max}$ was 141 mm for the calibration period and 140 mm for the validation period; annual precipitation was 1104 mm for the calibration period and 1143 mm for the validation period. We will add those numbers to the respective paragraph in methods next to the information about the increase in mean annual temperature by 0.7°C between both periods.

line 155 what is the range of threshold temperature throughout the catchments?

Threshold temperatures for individual catchments arising from median simulations (100 parameter sets) ranges from -1.58°C to 1.13°C. We will add this information to the respective part of the methods section.

Section 3.1 is not overly informative, in my opinion it can be moved to the supplement.

We think that showing the results for calibration and validation might be important for many readers to assess the overall ability of the model to simulate the individual components of the water cycle. Putting this part into supplement would probably cause a lot of readers to simply miss the information especially if this would be the only supplementary information. Therefore, we prefer to keep this part in the main text (unless there will be need for even more supplements).

Figure 4 (and Figure 8): catchments are sorted by "mean" elevation, also add an arrow and write elevation next to y axis, and at least give starting and end value (115m a.s.l. to 1602m a.s.l.)

We agree, we will indicate the elevation ranges in Figure 4 (and 8) and add "mean" into the figure caption.

Figure 4 (and Figure 6): make it clear, that you show the results for four specific catchments maybe by using the catchment names as headlines for the subpanels)

We agree, adding the catchments names to individual panels may increase the readability of both figures.

Figure 5 and Figure 7: make sure that you use different color coding, as you show different things (in Figure 5 Sf and in Figure 5 the regions)

Thank you for the suggestion. We will change the colour coding in Figure 7 to avoid confusions with Figure 5.

Figure 5 please mention the abbreviations (as in the axis titles) also in the figure caption

We agree, we will add the abbreviations to the figure caption.

Figure 7 is a bit confusing: In panel (a), do you show a point for each catchment where x is the mean of baseflow from all years having below average summer precipitation and y is the mean of baseflow from all years having below average SWEmax? If that is what I see in Figure 7a, than 58 out of 59 catchments have below average summer baseflow when they experience below average summer precipitation. However, only 40 out of 59 catchments had lower summer baseflow when having

lower SWE, which is not supporting your conclusion on the importance of SWE. Please revise this figure (and its caption) to make it clear what is shown.

With Figure 7, we wanted to show the relative importance of annual $SWE_{max}$ and summer precipitation on summer baseflow. For example, Figure 7a shows the median summer baseflow relative anomalies for years with below average summer precipitation (x axis) compared to the median of summer baseflow relative anomalies for years with below average $SWE_{max}$ (y axis). From Figure 7a it is clear that summer precipitation is more important for summer baseflow than $SWE_{max}$ (as we mentioned in line 277 of the original manuscript). Nevertheless, Figure 7b indicated that for the majority of catchments, the summer baseflow for years with below-average summer precipitation increased when there was simultaneously above-average $SWE_{max}$. Moreover, some of the catchments showed even positive summer baseflow anomalies for above-average $SWE_{max}$ despite negative anomaly of the summer precipitation. We are not saying that snow storages play a major role in generating summer baseflow, but results indicated that SWE is an important additional driver. An important implication for the future climate is that the summer baseflow might be lower because of lower snow storages even when summer precipitation would not change.

We agree that our explanation may not be fully clear. Therefore, we will consider reformulation (including changing the figure caption to be clearer).

In Figure 8 please consider using the same scale for the color bars to make the panels comparable.

Thank you for the suggestion. We considered using the same scale already during the manuscript preparation and decided in favour of different scales since the scales are of different magnitude. This is especially valid for panel (d) where the magnitude is of different order compared to other panels. However, we will try to standardize the scale for panels (a), (b) and (c), at least.

Discussion: You mention data errors in the headline of 4.1 but you did not discuss them.

Thank you for the notice. We will change the respective title to "HBV model setup and parameter uncertainty" to better describe the section content.

You need to better emphasize the challenges when separating liquid from solid precipitation within the HBV modelling framework. Maybe you can discuss the implications on your results a little more detailed.

The uncertainty of model parameters is discussed in the Section 4.1. We think it is an important topic since many of the model parameters might have an important effect on result interpretation. Nevertheless, a more detailed discussion of implications resulting from HBV parameterization was suggested also by Referee 1. Therefore, we will add more discussion on this topic.

The contribution from groundwater calculated with HBV is quite uncertain, you could also be looking at generally higher storage potential at higher elevations. Maybe you could consider discussing these uncertainties.

This comment also touches the issue mentioned by Referee 1. It is true that absolute values of groundwater storages simulated by the model may be uncertain since groundwater data were not used to calibrate the model (see also our response to Referee 1). Nevertheless, we were concentrated on relative differences (year-to-year variations) between groundwater fluxes in individual catchments rather than on absolute values. Some studies also showed that catchment storage calculated using different methods (water balance calculations, recession curve analysis, HBV modelling) are, in general, comparable and correlated, although the quantitative estimates may differ (Staudinger et al. 2017). The above study also showed that dynamic groundwater storage is correlated with elevation, indicating the relation of the groundwater storages and snow storages. The relative fraction of groundwater in streams and its sensitivity to inter-annual variations in snow storages was also shown by Carroll et al. (2019). Therefore, we will include more discussion on this topic to the revised version.

You mention a lot of interesting differences between the regions / catchments in the discussion, maybe you can add more information at an earlier part of the manuscript and build your story on these different regions.

Thank you for the suggestion. We agree that we should put more emphasis on regional consequences of our results because we are aware that our results are limited to the specific region and may not be easily generalized. We will add more information regarding catchments (e.g. by including a table with catchment characteristics as mentioned in one of your comments above) and we will consider reorganization of the discussion section to better highlight regional differences between our study catchments.

Conclusions: I'd appreciate if you could relate the statements with the according figures, that makes it easier for the reader to recap on where to find the evidence for the conclusions

Thank you for the suggestion. We will add some relevant links to results and figures.

The second objective (lines 86 & 87) is to show the importance of snowmelt "at different elevations", however elevation differences where not really mentioned and I also did not find any concluding remarks regarding this statement.

As we mentioned in one of the comments above, the dependence of individual characteristics on elevation is rather indirect and may not be easy to interpret although the elevation clearly influences the snowfall fraction and thus snow storages. We agree that mentioning the elevation as the most important catchment attribute might be confusing. We will reformulate both objectives and discussion to be clearer.

I am also not convinced that I saw results that support that "future liquid precipitation will not compensate the lower solid precipitation", please re-write or leave out.

With this sentence, we wanted to draw the attention on the fact that future decrease in snow storages might cause a decrease in annual runoff (even despite no changes in total amount of precipitation) and we think it is important to mention it. But maybe the formulation is not fully clear, so we will reformulate it to be clearer.

**References**

Carroll, R. W. H., Deems, J. S., Niswonger, R., Schumer, R. and Williams, K. H.: The Importance of Interflow to Groundwater Recharge in a Snowmelt-Dominated Headwater Basin, Geophys. Res. Lett., 2019GL082447, doi:10.1029/2019GL082447, 2019.

Staudinger, M., Stoelzle, M., Seeger, S., Seibert, J., Weiler, M. and Stahl, K.: Catchment water storage variation with elevation, Hydrol. Process., 31(11), 2000–2015, doi:10.1002/hyp.11158, 2017.

---

## Author Response (AR1)

**Authors' response to Referees comments**

Black text: Referee comment

Blue text: Authors' response

To make this final response easy to follow and clear, we used our previously published responses which we adjusted accordingly to what we exactly changed in the revised version.

**Response to the anonymous Referee #1**

We thank the reviewer for the valuable comments and suggestions to improve our contribution. We provide point-by-point reply below.

This paper investigates the role of snow (and rain) on streamflow across 59 Czech catchments. The objectives of the study are: to quantify how snow storages affect spring and summer runoff and to quantify how much runoff snowmelt in generates compared to rainfall. The study uses data of 50 catchments and simulations using the HBV model. They show the following results: 1. Snow runoff fractions exceed snowfall fractions (Fig 3), from which they conclude snow produces more runoff than rain. 2. How much runoff occurs in particular months varies between snow rich and snow poor years (Fig 4), with overall more runoff in snow rich years. 3. Several streamflow signatures vary between snow rich and snow poor years (Fig 5). 4. Summer base flow depends on both SWE and summer P (Fig 6+7) 5. That also in models annual and summer runoff strongly depend on the snow fraction (Fig 8).

These results are generally useful for the HESS' readership, as they address the important issue of how snow (and its anticipated future changes) affect river flow. However, before I can recommend publication of this article I think several things need to be addressed first:

- This HBV results suggest that snow produces runoff differently than rain. However, HBV treats snowmelt and rainfall largely similarly. This seems counterintuitive (or a paradox). It needs to become clearer in the modeling results how snowmelt is different than rain that leads to these runoff differences. Otherwise, I am not sure what we really learn from the presented results.

Thank you for this valuable comment. The changes in runoff due to snowfall/rain transition as simulated by modelling experiments (Fig. 8) pointed at two different aspects; 1) changes in annual water balance and 2) changes in seasonal runoff distribution. The first aspect was shown in Fig 8a and, in the model, was caused by lower actual evapotranspiration (AET) for higher snowfall fractions due to more days with snow cover (AET is calculated only for days with no snow cover on the ground in the model). We are aware that this particular result is influenced by the model structure which describes the whole rainfall-runoff process in a simplified way and thus the real catchment behaviour might not be captured correctly.

The second aspect, changes in seasonal distribution, was caused mainly by lower snow accumulation for lower snowfall fractions (more rain than snowfall) and by earlier snowmelt. This widely influenced the timing of groundwater recharge and thus spring and summer streamflow, low flows and deficit volumes (Fig. 8b-d). This

second aspect is, in our opinion, more important (although expected) since it widely influences the water availability during the warm period when the water demand is generally higher (for vegetation growth, agriculture, hydropower etc.).

We provided a better explanation in the abstract and introduction (better justification for knowledge gaps, L 10-11 and 81-82) and in discussion Section 4.4 (several places in the text between L 475-499). Some more information regarding the model structure and parameter uncertainty was added to Section 4.1 (L 375-385 of the revised version).

- The results listed above as 1-4 have all be shown before or are mostly trivial. There might be value in showing this again for the study catchments, but then I think the paper should better explain what we learn about the hydrology of these places, rather than largely use them as data for making some general statements.

We agree that most of the findings are not surprising as they mostly support our existing qualitative knowledge of how snow contributes to spring and summer runoff. However, we believe that the findings are still important even if they do not change our process understanding, and the quantification is a valuable and novel contribution. Besides, we were concentrated on a non-alpine (outside the Alps) region of Central Europe where there is only a little bit of published information on how ongoing changes in snow storages and snow/rain distribution at different elevations affect seasonal distribution of runoff. This is specifically important for identification of regions which might become more vulnerable to drought occurrence in the future. We also benefit from modelling approach enabling us to simulate both snow and rain runoff components and thus track the snow and rain signal in runoff. We believe that the fact that our focus was on the interplay of different rainfall-runoff components goes beyond what has been done before and thus it might bring some new insight into this topic.

Nevertheless, we are aware that our results are limited to the specific region and may not be easily generalized. Therefore, it was our intention to write the text in this respect. However, maybe, this was not always clear from our formulations. Therefore, we went through the text again to make it clearer, to better highlight the novelty and to put more emphasis on regional consequences of our results. More specifically, we added several statements regarding regional differences between catchments to different parts of the text in the results (text related to Fig. 7, L 289-306) and mainly discussion Section 4.3 (L 435-445 of the revised version). With this, we wanted to put more emphasis on regional aspects of our results. Nevertheless, more detailed investigation of differences between catchments and the potential influence of basin attributes on runoff generation has not been done in this study and needs to be further investigated (as mentioned in Section 4.2).

Additionally, we also tried to avoid drawing general conclusions since we are aware that our results should be related just to our study area or to an area with similar geography and climate. Therefore, we made minor edits at different places of the text in this respect.

- All results of groundwater recharge rely on the model output of an unvalidated flux (since no GW data are used). How do we have confidence they reflect actual groundwater recharge behaviours?

It is true that our results are not based on direct measurements of all individual runoff components since such measurements were not available for all analysed components. Therefore, we calibrated a model to simulate those components of the water cycle, for which observational data were not available. The HBV model, which was used in our study, is, despite its simplicity (and thus limiting ability to represent rainfall-runoff process in a fully physical way), widely used and accepted by the scientific community, especially for impact modelling at a

catchment scale. To better address the model uncertainty, we used an integrated multi-criteria approach to calibrate and validate the model using three objective functions reflecting both observed streamflow and SWE. The model allows using also groundwater (GW) data for calibration, but these data are not easily available for all study catchments. Besides, the density of the measuring network does not allow us to find the GW stations (either boreholes or springs as a proxy) which would sufficiently represent the whole catchment since the spatial variability in GW storages in a catchment is very large due to the variability in geology and soils. In contrast, the streamflow used to calibrate the model represents the integrated output from the whole catchment, and also the observed SWE data usually represents the catchment snow storage well enough (at least at a specific elevation zone). Therefore, it is questionable to which degree the use of the GW data for model calibration would result in more accurate simulations.

Additionally, we were concentrated on relative differences (year-to-year variations) between groundwater fluxes in individual catchments rather than on absolute values. Some studies also showed (Staudinger et al. 2017) that catchment storage calculated using different methods (water balance calculations, recession curve analysis, HBV modelling) are, in general, comparable and correlated, although the quantitative estimates may differ. Therefore, we believe that using HBV simulations for assessing the relative inter-annual differences in GW storage was an acceptable approach even when GW data were not used for model calibration.

We included the above explanation into discussion Section 4.1 (L 351-361) to better describe all uncertainties and limitations of using such a model for this topic.

- The paper contains a lot of unclear statements or language that if (interpreted as written) is wrong. I made a list of suggestions below, but this list is far from comprehensive. Please check the paper another time critically. This is really important because for too many statements it remains unclear what the authors claim to be true, and thereby makes it even impossible to review

Thank you for the comment. The text was corrected by a native speaker (hydrologist) and we carefully went through the text again to correct all language errors (including those mentioned below).

**Detailed comments**

L9: add the word "often" (or something similar), otherwise this general statement is false.

Added.

L11: (and in winter runoff). Not necessary to state, but maybe not bad to mention.

Changed to "winter to summer runoff".

L14: model output, not model performance.

Changed.

L15: the simulations are not "hypothetical" as they have been performed. I the paper intends to say something like "Hypothetical scenarios were modelled"

Changed to "hypothetical scenarios".

L19-20: "This was documented by [: : :] from snow to rain" This does not seem to be a logical statement. Maybe change the verb "documented"?

"Documented" replaced by "demonstrated",

L22: would "reduced" be more specific than "affected" and therefore more informative?

The word "affected" was used in the original manuscript because the baseflow was not reduced in all catchments (as explained in the next sentence). Therefore, we prefer to keep the sentence as is to avoid confusions, although it is less informative.

L29: "largely affects" seems a bit odd. Maybe "often affects" or "can affect".

Changed to "significantly" since the snow impact on runoff seasonality is really important and often substantial.

L30: "tend to occur" not "occurs".

Changed.

L32: "to increase [: : :] climate changes". Why mention "precipitation"? And reword "to increase in air temperature" to "to increasing air temperatures". (And probably make "climate changes" singular).

Changed.

L34: "during winter" may be an unnecessary (and sometimes wrong) specification here. In many mountain areas the shift from snow towards rain will be biggest in spring and fall (when temperatures are often near 0) compared to winter (when temperatures are generally below zero even when it gets a bit warmer)

We removed "during winter".

L34: "a rate of"? I do not think this makes sense here. Please check what is intended to be said here.

We changed "rate" to "proportion". The whole part is a definition of "snowfall fraction" (which is firstly used here) to avoid confusion about this term.

L35: would "reduced" be more specific than "affected" and therefore more informative?

Changed.

L37-38: I understand why you say "On the contrary" but this only makes sense by having the reader guess that this has an opposite effect on total streamflow generated (which you don't say, nor make it clear that this is what you're thinking about). Therefore I would try to reword this a little.

We removed the sentence since the mentioned information has no link to the previous information and, thus, we think it is redundant in this context.

L39-40: "Changes in [: : :] and occurs earlier". Or "Reduced snow accumulation, and earlier and slower snowmelt cause earlier and less groundwater recharge (Beaulieu et al., 2012; Foster et al., 2016)."

Changed, thank you for the suggestion.

L41: to "lower elevations" (make plural)

Changed.

L44-45: "Higher snowpack generates higher groundwater flow driven by snowmelt rates and thus contributes more to streamflow 45 (Barnhart et al., 2016)." Does not seem to be a logical statement. Do you mean

something like "Higher snowpack disproportionally feed groundwater leading to more to streamflow (Barnhart et al., 2016)"?

Changed, thank you for the suggestion.

L54: "were" seems redundant.

Removed.

L57: Thus "using" not "uses"

Corrected.

L96: Consider removing "the" at the start of the sentence.

The sentence was reworded.

L110-111: "For this, we used a bucket-type HBV model (Lindström et al., 1997) in its implementation, called HBV-light (Seibert and Vis, 2012)" This sentence is clear, but I would recommend to rephrase it. (E.g. remove 'in its implementation")

We wanted to mention that we used an HBV-light version of the model, which is the specific software implementation of the original HBV model. We slightly reworded the sentence.

L121: "Different weights were tested to achieve the best possible performance of the model" This seems somewhat vague and arbitrary. What made you choose the particular weight in the end (i.e. what made them the "best")?

Although we tested different weights, it is true that we did not use any consistent approach to find the best values of these weights. The testing was done just based on our experience with the model and based on the literature. Therefore, it is true that the choice was rather arbitrary, although it reflected the main purpose of the model use (accurate simulation of both high and low flows, water balance and snow storages). We reformulated the respective part to be clearer (L 129-131).

L133-134: "The similar procedures for model set-up and calibration was also used earlier in (Jenicek et al., 2018), although in different region." Fix the language of this sentence. For example by: something like "This procedure for model set-up and calibration was also used in Jenicek et al. (2018), although for different region".

Changed, thank you for the suggestion.

L137: the simulations are not "hypothetical" as they have been performed. I the paper intends to say something like "Hypothetical scenarios were modelled"

Changed.

L182: I am unsure what "simulated correctly" would really mean here. Do you mean "accurately simulated"?

Yes, we mean "accurately simulated". We changed it.

L192-200: It is unclear to me to what extent these results originate from snow being more effective in producing runoff than rain or whether this is because of the seasonal timing of precipitation (independent whether it's snow or rain).

We don't know whether we correctly understand this comment. The results here (Fig. 3) were analysed for individual hydrological years (1 Nov – 31 Oct) and thus any deviations from 1:1 line (Fig. 3a) should indicate, in our opinion, differences in annual water balance rather than differences in seasonal runoff distribution. We added more explanation to the revised version by emphasizing that the figure shows annual (not seasonal) water balances. Additionally, we reformulated the figure caption to be clearer.

L435: "This particular result proves that snow is more effective in generating catchment runoff compared to liquid precipitation" seems like an overly strong statement. Tone down the word "prove" and choose something like "indicates" or "suggests".

We agree, changed to "suggests".

L454-456: ". An understanding of potential model artifacts might be important : : :" is very vague. Can it be made more specific?

This conclusion referred to the issue about how the model structure could influence results discussed in Section 4.1. After consideration we decided to remove this sentence from concluding Section 5 since it is not much informative. Nevertheless, we significantly enriched related discussion Section 4.1 to provide the reader with more specific information about potential effects of individual values of model parameters on results.

**Response to the anonymous Referee #2**

We thank the reviewer for the valuable comments and suggestions to improve our contribution. We provide point-by-point reply below.

The paper presents how snow processes influence runoff generation in mountainous catchments in Czechia. The presented results are not novel, and similar things have been shown across different regions. However, the manuscript could still be a valuable contribution for the readership of HESS. The overall structure of the manuscript is quite clear, but inconsistent language makes the paper sometimes hard to follow, especially throughout the introduction and discussion. Below I suggest some changes that should be considered prior to publication.

At this point, I am not convinced by the conclusion that "snow is more effective in generating catchment runoff compared to liquid precipitation". First of all, it is not clear what I actually see in Figure 3: Did you plot the mean of both groups (snow rich and snow poor) for every catchment? Please add some information to make this clearer.

Thank you for the comment. As you correctly assume, individual points in Figure 3 represent mean snowfall fractions and snow runoff fractions for snow-poor and snow-rich years for individual catchments. The snow-rich years were defined as years with annual $SWE_{max}$ above the third quartile of the study period (1980-2014), while the snow-poor years represent years with annual $SWE_{max}$ below the first quartile of the study period. Therefore, each point represents a mean calculated from 8-9 annual values derived from ~35-year-long time series.

By Figure 3, we wanted to demonstrate to what degree the snowfall is important for runoff generation compared to rainfall in our study catchments. This was also assessed by runoff coefficients calculated separately for snowfall to snowmelt runoff and rain to rainfall runoff (values are not shown in the paper, but they are discussed in Section 3.2). The higher runoff fraction for snow-generated runoff is caused mainly by lower actual evapotranspiration during winter (see also our response to Referee 1). Additionally, in modelling experiments we showed that the transition of snowfall to rain caused changes in 1) annual water balance and 2) seasonal runoff distribution, affecting groundwater recharge and summer low flows (Fig. 8).

The above explanation was used to clarify the text at several places, specifically 1) we added a bit more explanation to methodological Section 2.4 (regarding snow runoff calculation in HBV, L 170-172), 2) to results Section 3.2 (better description of Figure 3, including rewording of the figure caption, L 205-210) and 3) we added more explanation to the related part of discussion Section 4.4 (consequences for seasonal runoff distribution and water availability, several edits within lines 475-499).

Second, I'd like to see the same calculations (Figure 3) with the absolute values for total snowmelt runoff and total snowfall precipitation as 26% (on average) of total runoff might still be less than 20% (on average) precipitation.

It is true that, in absolute values, snowfall represents a higher water volume than runoff from snow. By definition of the "effect tracking" algorithm in HBV (as described in Section 2.4), the source of the input water (precipitation) can be changed only by refreezing (from rain to snow), but this process is rather negligible in absolute terms. Therefore, snowfall will be always higher than snowmelt runoff over the defined longer period (losses from evapotranspiration will likely be always higher than changes of water source from rain to snow due to refreezing). Nevertheless, in few catchments it happened that few hydrological years showed higher

snowmelt runoff than snowfall in that year since part of this snowmelt contribution came from the previous year (probably thanks to a long catchment storage). The mentioned effect is certainly worth investigating further, but it goes beyond the scope of the current manuscript.

We added more text regarding this issue to the methods (Section 2.4) to better explain the partitioning of the rain and snowmelt runoff. Additionally, the above explanation was added to discussion Section 4.2 (L 395-399) to provide the reader with thorough explanation and interpretation. We also created a new figure according to your suggestion which shows absolute snowfall and snowmelt runoff values, but, finally, we decided not to include it since we thought it would not provide any new information. However, we are open to include it if requested.

Also the increasing trend with elevation in my opinion is not visible in the results. There needs to be further analysis (maybe cluster in elevation groups) to convince readers. I understand that some of these results are also supported by the HBV modelling. However, you need to more explicitly convince readers that snow vs. rainfall processes can be well separated in the current modelling setup.

You are right that we cannot draw a direct conclusion regarding the effect of elevation. The increase in the difference between snowfall fraction and snow runoff fraction is statistically significant for catchments with higher snowfall fraction (see Fig. 3b) rather than higher elevation. In general, the dependence on elevation is not directly evident from the results, although some other results in our study (e.g. Fig. 5) indicated the relation with snowfall fraction, which generally increases with mean catchment elevation. Although the snowfall fraction significantly increases with mean catchment elevation in our study catchments (Pearson correlation coefficient is 0.53, $p$ value<0.001), the correlation coefficient might be influenced by the fact that mean catchment elevation cannot describe the hypsography of individual catchments.

We clarified it better in the revised version of the manuscript by 1) adding the above explanation to discussion Section 4.3 (L 435-444) and by 2) toning down the interpretation regarding the effect of elevation, and highlighted the effect of snowfall fraction instead (see Fig. 3b and Fig. 5) wherever relevant in the Introduction, Results and Discussion sections.

A better characterization of the catchments (i.e., the runoff regimes, precipitation and runoff seasonality) is warranted. This will help to better emphasize why these results are valuable and why it might be useful to show the results for these specific study regions. To people who are not familiar with topography and hydroclimatology of Czechia it would be very helpful to have more "background" information on the study catchments. Please add a table with information on mean, max, min size, elevation, precipitation, temperature, discharge,: : :

Thank you for the suggestion. We added a new table (Table 1 in the revised version) showing the main catchment attributes and meteorological characteristics. We agree that this information might be useful for readers. Similar comment was also made by Referee 1.

Similarly to the comment made by Referee 1 and based on one of your comment below, we added several statements regarding regional differences between catchments to various parts of the text describing the results (text related to Fig. 7; L 289-306) and mainly to discussion Section 4.3 (L 435-447). With this, we wanted to put more emphasis on regional aspects of our results. Nevertheless, more detailed investigation into differences between catchments and the potential influence of basin attributes on runoff generation has not been conducted in this study and remains to be carried out in the future (as mentioned in Section 4.2).

Additionally, we also tried to avoid drawing general conclusions since we were aware that our results should be related just to our study area or to an area with similar geography and climate. Therefore, we made minor edits at different places of the text in this respect.

What are the main differences between the regions, and the four sample catchments? This is important to interpret the results afterwards (some of them are shown based on the different sample catchments). If I interpret the DEM correctly your highest peak is only 1602 m a.s.l., some of the catchments are far below 1000m in peak elevation, do they even have snowfall / accumulation every year? I find it difficult that, in the discussion section, you interpret the results based on the different regions, however they are not well characterized.

The four selected catchments represent different geographical regions, geology and elevations. We added more detailed information to the text (L 231-233). Individual catchments were also described through their attributes and climate characteristics in newly added Table 1.

Although the mean catchment elevation ranges only from 491 to 1297 m a.s.l, all catchments have the seasonal snowpack every year (mean $SWE_{max}$ for individual catchments ranges from 35 mm for lowest catchments to 664 mm for highest catchments). This information was added to Section 2.1 (L 101-104).

**Detailed comments**

line 98 you claim that the selection criterion is timeseries >35 years however in line 104 /105 you write that three catchments do have less data

It is correct that three catchments have shorter time series (by one or two years compared to the rest of catchments). We are aware that it may bring some inhomogeneity into results, but since the shortening is only one or two years, we decided to include those relatively snow-rich catchments in the analysis. We slightly reformulated the respective sentence (L 98-99).

line 125 although I tend to believe that annual precipitation, peak SWE did not change significantly it would be great to see this (maybe in a table in the supplementary)

Thank you for the comment. The mean $SWE_{max}$ was 141 mm for the calibration period and 140 mm for the validation period; annual precipitation was 1104 mm for the calibration period and 1143 mm for the validation period. We added those numbers to the respective paragraph in the methods (Section 2.2, L 135-138) next to the information about the increase in mean annual temperature by 0.7°C between both periods.

line 155 what is the range of threshold temperature throughout the catchments?

Threshold temperatures for individual catchments calculated from median simulations (100 parameter sets) range from -1.58°C to 1.13°C. We added this information to the respective part of methodological Section 2.4.

Section 3.1 is not overly informative, in my opinion it can be moved to the supplement.

We think that showing the results for calibration and validation might be important for many readers to assess the overall ability of the model to simulate the individual components of the water cycle. Putting this part into the supplement would probably cause a lot of readers to simply miss the information especially if this would be the only supplementary information. Therefore, we prefer to keep this sub-section in the main text.

Figure 4 (and Figure 8): catchments are sorted by "mean" elevation, also add an arrow and write elevation next to y axis, and at least give starting and end value (115m a.s.l. to 1602m a.s.l.)

We agree, we added the elevation ranges as y-axis labels to Fig. 4 and Fig. 8. We added "mean catchment elevation" to the figure captions.

Figure 4 (and Figure 6): make it clear, that you show the results for four specific catchments maybe by using the catchment names as headlines for the subpanels)

We agree, adding the catchments names to individual panels may increase the readability of both figures. We added the names to Fig. 4 and Fig. 6.

Figure 5 and Figure 7: make sure that you use different color coding, as you show different things (in Figure 5 Sf and in Figure 5 the regions)

Thank you for the suggestion. We changed the colour coding in Fig. 7 to avoid confusions with Fig. 5.

Figure 5 please mention the abbreviations (as in the axis titles) also in the figure caption

We agree, we added the abbreviations to the figure caption.

Figure 7 is a bit confusing: In panel (a), do you show a point for each catchment where x is the mean of baseflow from all years having below average summer precipitation and y is the mean of baseflow from all years having below average SWEmax? If that is what I see in Figure 7a, than 58 out of 59 catchments have below average summer baseflow when they experience below average summer precipitation. However, only 40 out of 59 catchments had lower summer baseflow when having lower SWE, which is not supporting your conclusion on the importance of SWE. Please revise this figure (and its caption) to make it clear what is shown.

By Figure 7, we wanted to show the relative importance of annual $SWE_{max}$ and summer precipitation for summer baseflow. For example, Figure 7a shows the median summer baseflow relative anomalies for years with below-average summer precipitation (x-axis) compared to the median of summer baseflow relative anomalies for years with below-average $SWE_{max}$ (y-axis). From Figure 7a it is clear that summer precipitation is more important for summer baseflow than $SWE_{max}$ (as we mentioned in line 277 of the original manuscript; L 296-297 of the revised version). Nevertheless, Figure 7b indicated that for the majority of catchments, the summer baseflow for years with below-average summer precipitation increased when there was simultaneously above-average $SWE_{max}$. Moreover, some of the catchments showed even positive summer baseflow anomalies for above-average $SWE_{max}$ despite the negative anomaly of summer precipitation.

We are not saying that snow storages play a major role in generating summer baseflow, but results indicated that SWE is an important additional driver. An important implication for the future climate is that the summer baseflow might be lower because of lower snow storages even when summer precipitation would not change.

We agree that our explanation may not be fully clear. Therefore, we reformulated the respective part of results Section 3.2 (L 289-306), and we added more discussion to Section 4.3 (L 445-449).

In Figure 8 please consider using the same scale for the color bars to make the panels comparable.

Thank you for the suggestion. We tested unified scales already during the manuscript preparation and decided in favour of different scales since the scales are of different magnitude. This is especially valid for panel (d)

where the magnitude is of different order compared to other panels. Therefore, we prefer to keep the respective figure as is (besides the modifications described earlier).

Discussion: You mention data errors in the headline of 4.1 but you did not discuss them.

Thank you for the notice. We changed the respective title to "HBV model setup and parameter uncertainty" to better describe the section content.

You need to better emphasize the challenges when separating liquid from solid precipitation within the HBV modelling framework. Maybe you can discuss the implications on your results a little more detailed.

The uncertainty of model parameters is discussed in Section 4.1. We think it is an important topic since many of the model parameters might have an important effect on the result interpretation. Nevertheless, a more detailed discussion of implications resulting from HBV parameterization was suggested also by Referee 1. Therefore, we added more discussion on this topic to Section 4.1 (L 368-385), specifically we extended the discussion related to model parameters $T_T$, $S_{FCF}$ and the calculation of AET. Besides, we newly added a bit of discussion related to the snow routine structure and its possible effect on the ability of the model to simulate snow storages, as newly tested by Girons Lopez et al. (2020) (currently discussed in HESSD).

The contribution from groundwater calculated with HBV is quite uncertain, you could also be looking at generally higher storage potential at higher elevations. Maybe you could consider discussing these uncertainties.

This comment also touches the issue mentioned by Referee 1. It is true that absolute values of groundwater storages simulated by the model may be uncertain since groundwater data were not used to calibrate the model (see also our response to Referee 1). Nevertheless, we were concentrated on relative differences (year-to-year variations) between groundwater fluxes in individual catchments rather than on absolute values. Some studies also showed that catchment storage calculated using different methods (water balance calculations, recession curve analysis, HBV modelling) are, in general, comparable and correlated, although the quantitative estimates may differ (Staudinger et al. 2017). The above study also showed that dynamic groundwater storage is correlated with elevation, indicating the relation of the groundwater storages and snow storages. The relative fraction of groundwater in streams and its sensitivity to inter-annual variations in snow storages was also shown by Carroll et al. (2019).

Therefore, we included more discussion on the ability of the model to simulate the groundwater storage (Section 4.1, L 351-361; see also the response to the respective comment of Referee 1). We also included a bit more discussion in Section 4.2 (L 396-399 and 420-423) related to the catchment storage.

You mention a lot of interesting differences between the regions / catchments in the discussion, maybe you can add more information at an earlier part of the manuscript and build your story on these different regions.

Thank you for the suggestion. We agree and we tried to put more emphasis on regional consequences of our results because we are aware that our results are limited to the specific region and may not be easily generalized. Please, see our answer to your related general comment above to see all changes we did in this respect.

We also thought about some reorganization of the discussion section to better highlight regional differences between our study catchments, but after consideration of all alternatives (and reflecting other changes we did in the discussion section), we decided to keep the structure of the section as it was in the original manuscript.

Conclusions: I'd appreciate if you could relate the statements with the according figures, that makes it easier for the reader to recap on where to find the evidence for the conclusions

Thank you for the suggestion. We added some relevant links to figures to the Conclusion section.

The second objective (lines 86 & 87) is to show the importance of snowmelt "at different elevations", however elevation differences where not really mentioned and I also did not find any concluding remarks regarding this statement.

As we mentioned in one of the comments above, the dependence of individual characteristics on elevation is rather indirect and may not be easy to interpret although the elevation clearly influences the snowfall fraction and thus snow storages. We agree that mentioning the elevation as the most important catchment attribute might be confusing. We reformulated both objectives and discussion to be clearer. Please, see our answer to your related general comment above.

I am also not convinced that I saw results that support that "future liquid precipitation will not compensate the lower solid precipitation", please re-write or leave out.

By this sentence, we wanted to draw the attention to the fact that the future decrease in snow storages might cause a decrease in annual runoff (even despite no changes in total amount of precipitation), and we think it is important to mention it. However, we reformulated the sentence to "
[revised manuscript text omitted]

---

## Referee Report (RR1)

Thank you for considering my comments on the previous version. Although still being "wordy" in some parts (i.e. introduction and discussion) the overall clarity of the manuscript improved by the extended description of Figures and improvements in the discussion. The updated catchment descriptors (Table 1) are much appreciated, also the better reference to differences between the catchments throughout the text as well as the clear mentioning of the role of SWE in the individual basins. The discussion was updated regarding mentioned weaknesses of the HBV modelling framework.

I am still not fully convinced by the conclusion regarding the importance of snow processes in these montane watersheds, however I also agree that the extra amount of work needed to carry out a convincing analysis considering all processes involved is far out of scope here. Nevertheless I would recommend (and highly appreciate) if the authors do their best to make the data accessible to the community as Czechia is not well represented in most available datasets yet.

Overall, I appreciate the applied changes to the manuscript and have no further objection to publication.